# LOGICALLY CONSISTENT LANGUAGE MODELS VIA NEURO-SYMBOLIC INTEGRATION

**Diego Calanzone**[*]
DISI, University of Trento
diego.calanzone@studenti.unitn.it

**Stefano Teso**
CIMeC & DISI, University of Trento
stefano.teso@unitn.it

**Antonio Vergari**
School of Informatics, University of Edinburgh
avergari@ed.ac.uk

## ABSTRACT

Current large language models (LLMs) are far from reliable: they are prone to generating non-factual information and, more crucially, to contradicting themselves when prompted to reason about relations between entities of the world. These problems are currently addressed with large scale fine-tuning or by delegating reasoning to external tools. In this work, we strive for a middle ground and introduce a loss based on neuro-symbolic reasoning that teaches an LLM to be logically consistent with an external set of facts and rules and improves self-consistency even when the LLM is fine-tuned on a limited set of facts. Our approach also allows to easily combine multiple logical constraints at once in a principled way, delivering LLMs that are more consistent w.r.t. *all* constraints and improve over several baselines w.r.t. a given constraint. Moreover, our method allows LLMs to extrapolate to unseen but semantically similar factual knowledge, represented in unseen datasets, more systematically. Code available at https://github.com/ddidacus/loco-llm.

## 1 INTRODUCTION

Developing reliable large language models (LLMs) and safely deploying them is more and more crucial, particularly when they are used as sources of knowledge (Petroni et al., 2019; Ji et al., 2023; Bommasani et al., 2021; Andriopoulos & Pouwelse, 2023). To do so, one would need LLMs to be *factual* (Wadden et al., 2020), i.e., agreeing on single facts that appear in a knowledge base (KB), and *logically consistent* (Li et al., 2019; Mitchell et al., 2022), i.e., being able not to contradict themselves or a KB when prompted to perform complex reasoning. It has been abundantly shown that training on large datasets for question answering (QA) (Tafjord & Clark, 2021) alone cannot meet these desiderata (Evans et al., 2021; Lin et al., 2021; Liu et al., 2023; Elazar et al., 2021).

Factuality and consistency are intimately related. Enforcing factuality alone generally boils down to fine-tuning an LLM on a large KB of atomic facts (Kassner et al., 2021). When predicting the truth values of these facts, several works try to enforce the simplest form of consistency: that the probability of a true fact shall be one minus the probability of its negation (Burns et al., 2022). More sophisticated heuristics are possible, e.g., contrastive fine-tuning on a large QA dataset by jointly optimizing for truthfulness of model answers (Liu et al., 2023). All these approaches require large KBs and more crucially are tailored towards specific logical constraints.

When it comes to self-consistency w.r.t. more complex reasoning scenarios, e.g., ensuring that LLMs can perform modus ponens reasoning without contradicting themselves (Tafjord et al., 2022; Mitchell et al., 2022), one line of research focuses on employing *external* reasoning tools such as MAX-SAT solvers (Battiti, 2009) at inference time (Mitchell et al., 2022; Jung et al., 2022; Kassner et al., 2023). However, these approaches depend on the constant availability of a reasoner (and

---

[*]correspondence to, 🎲 = shared supervision

Figure 1: **Pipeline of our Logically Consistent (LoCo) LLMs.** LoCo-LMs are made factual and logically (self-)consistent by fine-tuning a base LLM according to a knowledge base of facts and rules (Section 3). The constraints $\alpha_i$ – which can be arbitrary propositional logic formulas – are compiled into a circuit (just once) and then used to encourage the model to allocate non-zero probability only to factual and consistent facts via a semantic loss (Xu et al., 2018), see Equation (SL).

sometimes also of a natural language inference model (Mitchell et al., 2022)) which can increase the cost of inference for every reasoning step. At the same time, training the LLM to reason is not possible or hindered by the hardness of backpropagating through the solver (Pogancic et al., 2020).

In this work, we show how to improve factuality and self-consistency of LLMs without external components by leveraging recent advancements in neuro-symbolic (NeSy) reasoning (De Raedt et al., 2021). This is done by turning complex reasoning tasks into logical constraints that can be compiled into computational graphs (Vergari et al., 2021). Specifically, we fine-tune an LLM by a principled objective: maximising the *exact* probability of a constraint to hold, which goes under the name of *weighted model counting* (Chavira & Darwiche, 2008) in probabilistic reasoning or *semantic loss* (Xu et al., 2018) when used as a regularizer in deep learning (Zhang et al., 2023; van Krieken et al., 2024). This in turn encourages the LLM to perform exact probabilistic reasoning at training time by maximising the probability of beliefs that comply with the provided set of constraints.

We empirically show how given incomplete factual knowledge, e.g., by providing only a limited number of known facts, the LLM can learn truth beliefs for new facts while keeping logical consistency w.r.t. prior knowledge. Moreover, our approach is agnostic to the logical constraints considered and can deliver a single training objective that can improve multiple consistency scores at once. In our experiments, with a single offline training session, LLMs trained with our objective outperform models relying on external solvers, and are more factual and logically consistent in low-data regimes when compared to standard supervised fine-tuning over KBs of facts.

**Contributions.** Summarizing, we: i) introduce **Lo**gically-**Co**nsistent LLMs (LoCo-LMs), a novel and principled fine-tuning strategy designed to improve factuality and (self-)consistency of LLMs based on probabilistic NeSy reasoning (Section 3), and ii) we rigorously evaluate the ability of LoCo-LMs to improve self-consistency w.r.t. several reasoning scenarios – when fine-tuned for certain constraints and evaluated over others – without hurting fluency (Section 5).

## 2 CONSISTENCY THROUGH THE LENSES OF PROBABILISTIC REASONING

We formalize the different reasoning scenarios we would like an LLM to be (self-)consistent with, and highlight the shortcomings of commonly used LLMs when prompted to reason in this way.

**Factuality.** We view a pre-trained LLM as a collection of truth beliefs about facts over which it can *reason*. The simplest reasoning task is ***factual reasoning***, i.e., determining the veridicity of a fact. For example, consider the fact $f$ in textual form "an albatross is a bird". It can be commonly encoded in knowledge bases (KBs) such as BeliefBank (Kassner et al., 2021) as a (*subject-relation*, *property*) pair, for instance, (albatross-is, bird). To inspect whether an LLM believes a fact to be true, we can prompt it with a question like "Is an albatross a bird?", the LLM can supply a binary prediction of the form "Yes"/"No" or "True"/"False",[1] encoding its belief that the fact $f$ holds or not. Therefore, given an LLM modeling a parameterized distribution $p_\theta$, we can consider the probability of generating a token $x_t$ encoding a binary answer, according to $p_\theta$, after observing the token sequence $x_1, \ldots, x_{t-1}$ encoding the question about the fact, to be the probability of the LLM believing that the truth value $z_f$ of fact $f$ is either true ($\top$) or false ($\bot$). That is, for true facts,

$$p_\theta(z_f = \top) = p_\theta(x_t = \ell_{\text{true}} \mid x_1, \ldots, x_{t-1} = \text{"Is an albatross a bird?"}) \qquad (1)$$

---

[1] We note that such an answer can be highly dependent on the format of the prompt. For this reason, in our experiments we use several prompts, whose format is detailed in Section 5.

where $\ell_{\text{true}}$ is an affirmative token, e.g., one among "yes", "true", "Y", "T", etc. Analogously, we can compute $p_\theta(z_f = \bot)$ by checking if the LLM answers a token $\ell_{\text{false}}$ is "no", "false", "N", "F", etc. To determine the model's belief, we query [2] the most likely next token $\hat{x}_t$ and check whether it falls in $\ell_{\text{true}}$ or $\ell_{\text{false}}$, and set it to "undetermined" if it falls into neither.

Given an external KB, we say an LLM is *factually consistent*, or simply factual, w.r.t. a fact $f$ in the KB with truth value $z_f^*$, if its answer (mapped to a truth assignment as described above) matches $z_f^*$, and factually inconsistent otherwise. [3] This perspective leads to interpreting factual reasoning as a binary question answering (QA) task (Burns et al., 2022; Kassner et al., 2021; Mitchell et al., 2022). From Equation (1), one can see that a simple strategy to make an LLM more factual is that of minimizing the cross-entropy (XENT) of $p_\theta$ over an external KB containing training questions with ground truth answers. We compare against it in our experiments (Section 5).

**Negation consistency.** While effective for many QA scenarios (Liu et al., 2023; Tian et al., 2023), increasing factual consistency by XENT minimization does not prevent the LLM from being logically inconsistent under other simple constraints, e.g., contradiction (Kassner & Schütze, 2019; Cohen et al., 2023; Jang & Lukasiewicz, 2023). Given a textual representation for a fact $f$, e.g., "an albatross is a bird", and another one $\widetilde{f}$ encoding its negation, e.g., "an albatross is *not* a bird", we say *negation self-consistency* holds if

$$z_f \oplus z_{\widetilde{f}} \iff (z_f \wedge \neg z_{\widetilde{f}}) \vee (\neg z_f \wedge z_{\widetilde{f}}), \tag{Neg}$$

where $\oplus$ denotes the logical operator XOR. In other words, we would like an LLM to consistently answer either affirmatively or negatively when asked about the truth of a statement and its negation. Negation consistency is very challenging for LLMs (Kassner & Schütze, 2019; Elazar et al., 2021; Jang & Lukasiewicz, 2023). For example, in our experiments LLaMa-2 70b (Touvron et al., 2023) answers "true" to both questions "Is an albatross an organism?" and "Is an albatross not an organism?". From a probabilistic perspective, a simple sufficient condition for negation consistency is that $p_\theta(z_f = \top) = 1 - p_\theta(z_{\widetilde{f}} = \top)$. This is hard to be systematically guaranteed and in practice has been addressed by applying ad-hoc heuristics during fine-tuning (Burns et al., 2022), which however cannot be exploited to enforce consistency to other constraints, such as implication, discussed next.

**Implication consistency.** Given two textual representations of facts $f_1$ (antecedent, e.g., "an albatross is a bird") and $f_2$ (consequent, "an albatross is an animal") we say that the first implies the second if it holds that

$$(z_{f_1} \to z_{f_2}) \iff (\neg z_{f_1} \vee z_{f_2}). \tag{Imp}$$

As with factuality, consistency (resp. self-consistency) holds if the answers of the LLM to a prompt satisfy the truth values according with the above implication and an external KB (resp. the inner beliefs of the LLM). Furthermore, letting $z_{f_1}^*$ be the truth value of $f_1$ recorded in the KB, we can define a *factual variant of the implication* that restricts the constraint to take $z_{f_1}^*$ into account, that is, when the LLM is prompted about $f_2$, it should derive its truth value $z_{f_2}$ according to

$$(z_{f_1} = z_{f_1}^*) \wedge (z_{f_1} \to z_{f_2}) \tag{F-Imp}$$

This can be seen as a relaxation of classical modus ponens reasoning (Robinson & Voronkov, 2001). While simpler to capture from text corpora, implication consistency can still be challenging for LLMs (Kassner et al., 2023; Yang et al., 2024). For example, given the rule $f_1 \to \neg f_2$, where $f_1$: "an albatross is an animal" and $f_2$: "an albatross is a virus", we wish the LLM to answer with "Yes" and "No" respectively, which maps to the truth assignment $z_{f_1} = \top$, $z_{f_2} = \bot$. LLaMa-2 70b violates the provided rule with the inconsistent belief, $z_{f_2} = \bot$, i.e. "an albatross is a virus" is labeled as a true statement.

**Reverse implication consistency.** Equation (Imp) is logically equivalent to $\neg z_{f_2} \to \neg z_{f_1}$, nevertheless an LLM that is logically consistent w.r.t. the implication of $f_1$ over $f_2$ it might not necessarily be consistent w.r.t. the implication of $\widetilde{f_2}$ over $\widetilde{f_1}$, representing the negation of $f_2$ and $f_1$ respectively.

---

[2] We keep a default temperature $t = 1.0$. Dropout is disabled to generate outputs systematically.

[3] Similarly, an LLM is *factually self-consistent* w.r.t. $f$ if it answers in the same logically consistent way (e.g., $z_f$ is always $\top$) when asked to answer the same or semantically equivalent prompts different several times. Since this is harder to measure – as it strongly depends on the sampling strategy – we focus on factual consistency only.

For example, while LLaMa-2 70b is logically consistent w.r.t. $z_{f_1} \to z_{f_2}$ with $f_1$ : "an albatross is an organism", $f_2$ : "an albatross is a living thing", it violates $z_{\widetilde{f_2}} \to z_{\widetilde{f_1}}$ as it classifies $z_{\widetilde{f_2}}$ : "an albatross is not a living thing" as false but $z_{\widetilde{f_1}}$ : "an albatross is not an organism" as true. Furthermore, an LLM that is logically consistent w.r.t. reverse implication and factual w.r.t. a KB should satisfy

$$(z_{\widetilde{f_2}} = \neg z^*_{f_2}) \wedge (z_{\widetilde{f_2}} \to z_{\widetilde{f_1}}) \qquad \text{(REV-F-IMP)}$$

where $\neg z^*_{f_2}$ indicates the opposite of the truth value stored in the KB for $f_2$. This factual reverse implication scenario can be thought as a relaxation of *modus tollens* (Robinson & Voronkov, 2001).

**More complex constraints.** As just discussed, constraints such as negation, logical implication and reverse implication already pose challenges to state-of-the-art LLMs in terms of consistency. While we will focus on the Llama 2 LLM family in this work, similar shortcomings have been highlighted for even larger models such as ChatGPT and GPT-4 (Jang & Lukasiewicz, 2023). Nevertheless, they constitute only a small fraction of the possible real-world reasoning scenarios LLMs can be asked to deal with. Consider for example the following textual representations of facts, as extracted from EntailmentBank (Dalvi et al., 2022): $f_1$ : "melting is a kind of phase change", $f_2$ : "the ice melts", $f_3$ : "the ice undergoes a phase change", $f_4$ : "phase changes do not change mass", $f_5$ : "the mass of the ice will not change". They obey the following logical constraint

$$(z_{f_1} \wedge z_{f_2} \to z_{f_3}) \wedge z_{f_4} \to z_{f_5}. \qquad (2)$$

In the next section, we will introduce our general framework that can improve logical consistency of fine-tunable LLMs w.r.t. *any* logical constraint expressible in propositional logic.

## 3 LOGICALLY-CONSISTENT LLMS VIA NESY INTEGRATION

We assume we are given a KB comprising a set of textual statements and associated truth values $\mathcal{D}_F = \{(f_1, z^*_{f_1}) \ldots, (f_n, z^*_{f_n})\}$, encoding simple facts such as "an albatross is a bird" (true) and "a computer is a bird" (false), and a set of logical constraints $\mathcal{D}_C = \{\alpha_1, \ldots, \alpha_m\}$ – e.g., implications, negations or more complex constraints like those defined in Section 2 – over the facts in $\mathcal{D}_F$.

Given a pre-trained LLM encoding a distribution $p_\theta$ over tokens, our objective is to fine-tune it to be more consistent w.r.t. $\mathcal{D}_F, \mathcal{D}_C$ and itself. As an important side benefit, we expect the fine-tuned LLM to generalize to – and be consistent with – the truth values of unseen facts $f_{n+1}, f_{n+2}, \ldots$, that can be either logically inferred by applying the constraints in $\mathcal{D}_C$ to $\mathcal{D}_F$ (e.g., by applying modus ponens) or that are semantically similar to facts in $\mathcal{D}_F$. E.g., since albatross and cockerel are semantically similar for an LLM, we expect an LLM consistent with the constraint "an albatross is a bird" $\to$ "an albatross can fly" to correctly infer that "a cockerel can fly" too.

A principled probabilistic approach to do so is to encourage the LLM $p_\theta$ to allocate all probability mass to configurations of truth values that are consistent with the constraints $\alpha_i \in \mathcal{D}_C$, for instance by penalizing it proportionally to the probability it allocates to inconsistent truth values for all facts in the KB. For every $\alpha_i$, the total probability allocated to the consistent configurations is

$$\Pr(\alpha_i) := \mathbb{E}_{\mathbf{z} \sim p_\theta(\mathbf{z})}[\mathbb{1}\{\mathbf{z} \models \alpha_i\}] = \sum_{\mathbf{z} \models \alpha_i} p_\theta(\mathbf{z}) \qquad (3)$$

where $\mathbf{z}$ is a vector containing the truth assignments $z_1, \ldots, z_K$ of all the $K$ facts appearing in the constraint $\alpha_i$, and $\mathbf{z} \models \alpha_i$ indicates that the assignment $\mathbf{z}$ satisfies the constraint, and the individual probabilities $p_\theta(\mathbf{z})$ are obtained using Equation (1). For example, consider two facts $f_1$ : "a daffodil is a flower" and $f_2$ : "a daffodil is mortal" and the constraint $\alpha' : z_{f_1} \to z_{f_2}$ stating that being a flower entails that the daffodil is mortal. Then, all the configurations of $\mathbf{z} = (z_{f_1}, z_{f_2})$ would satisfy $\alpha'$ with the exception of $(\top, \bot)$ which clearly violates it. Equation (3) is a special instantiation of computing the weighted model count (WMC) (Chavira & Darwiche, 2008; van Krieken et al., 2024) of a logical formula $\alpha_i$, where the weights associated to each model (a satisfying assignment to the formula) are given by the probabilities encoded by the LLM.

Furthermore, we can rewrite such probabilities $p_\theta(\mathbf{z})$ as the product the probabilities of the truth values of each fact, noting that for many LLM architectures they are conditionally independent given the embeddings at the last layer. By taking the logarithm and reversing it into a minimization problem, we obtain the *semantic loss* (SL) (Xu et al., 2018) objective that our LoCo-LMs minimize:

$$\mathcal{L}(\alpha_i, p_\theta) = -\log \sum_{\mathbf{z} \models \alpha_i} \prod_{j:\mathbf{z} \models z_{f_j}} p_\theta(z_{f_j}) \prod_{j:\mathbf{z} \models \neg z_{f_j}} (1 - p_\theta(z_{f_j})) \qquad \text{(SL)}$$

where $j : \mathbf{z} \models z_{f_j}$ (resp. $j : \mathbf{z} \models \neg z_{f_j}$) indicates that the $j$-th fact in $\alpha_i$ is associated $\top$ (resp. $\bot$). Consider the implication constraint $\alpha'$ as defined before for encoding that a daffodil is mortal for being a flower. Its satisfying assignments are $\mathbf{z} \models \alpha' \in \{(\top, \top), (\bot, \top), (\bot, \bot)\}$. Then, the summation in Equation (SL) amounts to computing:

$$p_\theta(z_{f_1} = \top)p_\theta(z_{f_2} = \top) + (1 - p_\theta(z_{f_1} = \top))p_\theta(z_{f_2} = \top) + (1 - p_\theta(z_{f_1} = \top))(1 - p_\theta(z_{f_2} = \top)))$$

where we can obtain the individual probabilities of facts being true directly by reading off the likelihood of utterances produced by the LLM, that is:

$$p_\theta(z_{f_1} = \top) = p_\theta(x_t = \ell_{\mathsf{true}} \mid x_1, \ldots, x_{t-1} = \text{"Is a daffodil a flower?"})$$
$$p_\theta(z_{f_2} = \top) = p_\theta(x_t = \ell_{\mathsf{true}} \mid x_1, \ldots, x_{t-1} = \text{"Is a daffodil a mortal?"}).$$

In the case of a constraint such as Equation (F-IMP), the inner summation of the SL would reduce to a single configuration $\mathbf{z} = (\top, \top)$ when $z_{f_1}^* = \top$, which can be interpreted as a special kind of cross-entropy computed only on pairs of facts considered to be jointly true in the KB, and to the set $\{(\bot, \top), (\bot, \bot)\}$ when $z_{f_1}^* = \bot$. Note that Equation (SL) is *agnostic to the kind of logical constraint involved*, and therefore makes our approach general enough to tackle several settings where consistency-preserving solutions have been devised for specific constraints (Burns et al., 2022; Kassner et al., 2023; Mitchell et al., 2022).

Crucially, the procedure to compute the models of a logical constraint can be automated. Now, naively computing the sum in Equation (SL) would require exponential time w.r.t. the number of possible facts in $\mathbf{z}$. In fact, computing the WMC of a logical formula is a #P-hard problem in general (Chavira & Darwiche, 2008). However, thanks to recent advancements in neuro-symbolic reasoning, we can compute that probability and differentiate through it efficiently (Darwiche, 2011; Xu et al., 2018; Ahmed et al., 2022a). Specifically, we rely on modern *compilers* that translate a logical formula $\alpha_i$ into compact and differential computational graphs called *circuits* (Darwiche, 2003; Vergari et al., 2019), such as sentential decision diagrams (Darwiche, 2011; Oztok & Darwiche, 2015; Choi & Darwiche, 2013), cf. Appendix A for details. In our scenario, compilation is extremely fast taking only 2.5 milliseconds to compile a single logical formula and compute the loss on BeliefBank (Section 5).

To recap (cf. Figure 1), during training we loop over every constraint in $\alpha_i \in \mathcal{D}_C$, prompt the LLM to gather the probabilities of every fact participating in $\alpha_i$ to be true and plug them in our only loss, as described in Equation (SL). Then, we backpropagate as to fine-tune (some of) the parameters $\theta$ of the LLM, by using LoRA (Hu et al., 2021) and quantization (Dettmers et al., 2023) if necessary. This simple and principled recipe is able to scale well and is extremely effective at improving logical consistency on a number of well-known benchmarks, as discussed in Section 5.

## 4 RELATED WORK

**LLMs and factual reasoning.** LLMs are increasingly being employed as implict KBs (Petroni et al., 2019; AlKhamissi et al., 2022), however ensuring they are factually consistent is still an open challenge (Wang et al., 2023; Augenstein et al., 2023). A number of works augment LLMs with external KBs, especially in the context of QA, and with the primary aim of improving answer factuality (Kassner et al., 2023; Mitchell et al., 2022; Li et al., 2024b). A popular approach to do so is retrieval augmented generation (Lewis et al., 2020; Li et al., 2024a), which however is not yet suited for more complex reasoning scenarios. Alternatively, external KBs have been used to improve reasoning, e.g., via prompt learning (Palagin et al., 2023) or ex-post model editing (Shi et al., 2023). However, current knowledge editing methods, including supervised fine-tuning, do not guarantee the propagation of factuality between units of knowledge related by logical relations (Cohen et al., 2023; Akyürek et al., 2024). Mitigating hallucinations in LLMs (Andriopoulos & Pouwelse, 2023; Rawte et al., 2023) is related to enforcing factuality, but as generated inconsistencies might not map to a single entry in a KB, they are harder to detect and prevent (Hong et al., 2024).

**More complex reasoning with LLMs.** Much less attention has been posed to composite forms of reasoning, e.g., combining modus ponens and consistent negation. Even when this is done, reasoning is generally cast as a QA task, where an LLM has to predict the satisfiability of logical formulas of different complexities, as in benchmarks such as SimpleLogic (Zhang et al., 2022) or LogicBench

(Parmar et al., 2023). Implication or entailment (MacCartney, 2009; Evans et al., 2018) are also usually cast as a QA prediction task (Raj et al., 2023). BeliefBank (Kassner et al., 2021) provides collections of implication constraints to test this, while more sophisticated benchmarks such as EntailmentBank (Dalvi et al., 2022) include more complex implications, e.g., trees of natural language statements. Shortcomings in consistent reasoning have been recently highlighted for larger LLMs such as ChatGPT and GPT-4 variants (Jang & Lukasiewicz, 2023), which are however harder to fine-tune efficiently. Other works (Berglund et al., 2023) highlighted how (even large) LLMs suffer from not being able to recognize the logical equivalence of "A is-a B" and "B is-a A" relationships.

For complex reasoning scenarios, logical consistency can be improved in a number of ways, the most successful of which involves external tools, such as MaxSAT solvers, which flip the predictions of an LLM to be (approximately) consistent with a set of related questions, as done by ConCoRD (Mitchell et al., 2022) and maieutic prompting (Jung et al., 2022). Analogously, self-consistency can be ameliorated by first constructing a belief graph – a factor graph relating the beliefs of an LLM fine-tuned on implications such as Entailer (Tafjord et al., 2022) – over which a MaxSAT solver is applied (Kassner et al., 2023). Higher level constraints can also be checked and enforced with external verifiers (Wang et al., 2024). Differently from LoCo-LMs, backpropagating through these tools is hard (Pogančić et al., 2019). Moreover, while they can guarantee self-consistency among facts *within* every call to a solver, this cannot be done for the same facts *across* different calls.

**Semantic loss & other NeSy approaches** There is a vast literature on NeSy integration methods (De Raedt et al., 2019; 2021), most of which are used for enforcing constraint on tabular data (Giunchiglia & Lukasiewicz, 2020), image data (Xu et al., 2018; Shindo et al., 2021; Ahmed et al., 2022a) and more recently video recognition (Giunchiglia et al., 2023) with the purpose of building trustworthy predictors. Several variants of the semantic loss (Xu et al., 2018; Ahmed et al., 2022b; 2024) and neural weighted model counting (van Krieken et al., 2024) have been proposed. Closer to our work, Zhang et al. (2023) applied a semantic loss to instill first-order rule constraints in the embedding space of entities in encoder-only models to reason on the CLUTTR benchmark (Sinha et al., 2019), comprising semi-synthetic stories involving hypothetical families. Richardson & Wijnholds (2024) propose to combine LLMs and a semantic loss for consistency analogous to ours. Faghihi et al. (2023) *approximate* a semantic loss via sampling to improve consistency of *only* implications for small BERT-like models. We do not need approximations as we rely on exact computations via compilation while scaling to larger constraints and combining different constraints together. Fuzzy logic (van Krieken et al., 2022) can be used to distill regularizers that can promote consistency (Li et al., 2019). Differently from our probabilistic logic approach however, they are syntax-dependent, i.e., rewriting a constraint into a logically equivalent one would yield a different penalty term and can greatly influence optimization (Di Liello et al., 2020).

## 5 EXPERIMENTS

We aim to answer the following research questions: **RQ1**: Can LoCo-LMs achieve comparable or superior consistency to methods using external reasoners using less training data? **RQ2**: Can LoCo-LMs retain good consistency to unseen types of constraint at training time? How much does training on all the constraints jointly improve consistency overall? **RQ3**: Can LoCo-LMs transfer consistent knowledge to domains out of the training distribution?

### 5.1 RQ1: HOW DO LoCo-LMs PERFORM COMPARED TO EXTERNAL SOLVERS?

We reproduce the experimental setting of Mitchell et al. (2022) to compare against ConCoRD, a symbolic layer that uses a MaxSAT solver to impose self-consistency for implication ex-post. Maieutic prompting employs essentially the same strategy (Jung et al., 2022).

**Data.** We train LoCo-LMs on the BeliefBank (Kassner et al., 2021). We use the three splits as in Mitchell et al. (Mitchell et al., 2022): a "calibration" set of $1,072$ annotated facts about 7 entities of the form *(subject, property, true/false)* used for training, a "silver" set of $12,636$ facts about 85 entities used for evaluation, and a set of 2224 valid abstract logical implications. We generate ground implication rules ($\mathcal{D}_C$) by looking up the subjects of all facts in the training set: if the antecedent or the consequent fact of the general constraint is known for that subject, we add the subject ground implication constraint to the dataset. Appendix B.1.1 details the whole process.

Table 1: **LoCo-LMs achieve better logical self-consistency and factuality than ConCoRD** (Mitchell et al., 2022) as measured via Equation (4) and $F_1$ scores when fine-tuned only on T1 facts only and boost performance in the presence of a small fraction of T1+T2 facts (5-10%). A similar trend is visible on training data (Appendix B.1.1).

| METHOD | TRAIN SUBSET | ANT $F_1$ | CON $F_1$ | TOT $F_1$ | IMP |
|---|---|---|---|---|---|
| CONCORD | | | | 0.91 | 0.91 |
| MACAW-LARGE | | 0.52 | 0.90 | 0.81 | 0.83 |
| MACAW+XENT | T1 | 0.13 | 0.01 | 0.03 | 0.72 |
| LoCo-MACAW | T1 | **0.79** | **0.98** | **0.96** | **0.99** |
| MACAW+XENT | T1+T2 (5%) | 0.23 | 0.78 | 0.72 | 0.82 |
| LoCo-MACAW | T1+T2 (5%) | **0.67** | **0.83** | **0.81** | **0.92** |
| MACAW+XENT | T1+T2 (10%) | **0.55** | **0.97** | **0.91** | 0.90 |
| LoCo-MACAW | T1+T2 (10%) | 0.45 | **0.97** | 0.89 | **0.93** |
| MACAW+XENT | T1+T2 (75%) | **0.85** | **0.99** | **0.97** | **0.98** |
| LoCo-MACAW | T1+T2 (75%) | 0.79 | **0.99** | 0.95 | **0.98** |

To measure generalization across entities, we generate two controlled splits of the training calibration set: *T1 facts*, appearing either as antecedents or consequents in the constraints; *T2 facts*, appearing exclusively as consequents. The goal is to correctly guess the consequents by seeing only the antecedents and the constraints. We subsequently test the effects of pure supervised fine-tuning on a portion of random facts from the whole calibration set (T1+T2).

**Models.** As in Mitchell et al. (2022), we use Macaw-Large (Tafjord & Clark, 2021) ($770M$ parameters), a sequence-to-sequence language model capable of multi-angle QA with fixed prompt templates. We keep the same prompts used for Macaw, reported in Appendix F.1. At test time, we verify the validity of the answer format and consider any invalid or negative response as a belief with label "false". We adopt a similar set of hyperparameters as for Macaw (Tafjord & Clark, 2021): we fine-tune our models for 3 epochs with a learning rate fixed to $\gamma = 3 \cdot 10^{-4}$, batch size 4 with gradient accumulation (64/16 steps), on one nVidia A30 24GB GPU. We use AdamW (Loshchilov & Hutter, 2016) as optimizer with a default weight decay $\lambda = 10^{-2}$.

**Competitors and Metrics.** We compare ConCoRD as applied to Macaw-Large, using RoBERTa-ANLI (Liu et al., 2019) for relationship inference, versus a pre-trained Macaw-Large model from Tafjord & Clark (2021) as zero-shot baseline and our LoCo version of it (LoCo-Macaw). We evaluate our models for *factuality* and *implication self-consistency*. We measure the former with the $F_1$ score to account for the unbalance between false and true facts (Kassner et al., 2021). Factuality is measured on the two splits (antecedents and consequents) and the complete facts set (Tot) for both calibration and silver splits. For ***implication self-consistency***, sometimes named just "consistency" (Li et al., 2019), we query beliefs from LLMs about the complete facts set and count the fraction of violated constraints in $\mathcal{D}_C^{\text{test}}$ according to the implication rule (IMP), that is, when a true antecedent for the model implies a false consequent, to then compute:

$$1 - |\{\alpha_i = (z_j \to z_k) : z_j = \top, z_k = \bot\}| \ / \ |\{\alpha_i = (z_j \to z_k) : z_j = \top\}|. \tag{4}$$

**Results.** Table 1 reports all metrics for all models. We firstly observe a net improvement in both factuality and logical consistency with our LoCo-LMs, compared to pre-trained Macaw-Large and the ConCoRD variant. Standard supervised fine-tuning with the XENT loss on antecedent facts is insufficient: due to a class imbalance between true facts ($\sim 10\%$) and false facts ($\sim 90\%$), the model tends to label any statement as "false". This is accentuated in the training distribution (see Appendix B.1.1). Assuming the language model can access to a portion of consequent facts, LoCo-LMs still yields better logical consistency and factuality for unseen consequents in low-data regimes (e.g., 5-10% of the T1+T2 dataset) compared to canonical supervised fine-tuning. When they are allowed to see more data (e.g., 75% of the T1+T2 dataset), traditionally fine-tuned models can "cheat" and directly learn about the consequents (somehow equivalent to memorizing a single row of the truth table). In this scenario, LoCo-LMs achieve comparable logical self-consistency and factuality over consequents, but less on the antecedents.

In conclusion, we observe ***our fine-tuning method allows Macaw-large to be more logically self-consistent than with an external solver***. We conjecture that this is possible thanks to the high semantic similarity between facts in the train and test splits (Appendix E.1). In terms of inference speed, our LoCo-LMs take less time that querying the same base model and an additional reasoner[4], at the cost of a one-time training step that can be amortized. Moreover, our semantic loss

---

[4]On BeliefBank, LoCo-LMs take 2405.28s at test time, compared to ConCoRD, 3669.33s.

is more sample-efficient than XENT fine-tuning to achieve higher logical consistency especially with small portions of ground-truth data. *We point out that our* LOCO-LMS *can be combined with external solvers at inference time, improving even more (self-)consistency.*

## 5.2  RQ2: HOW DO LOCO-LMS DEAL WITH DIFFERENT LOGICAL CONSTRAINTS?

**Setting.** As in Section 5.1, we use BeliefBank to train and evaluate LOCO-LMS on different types of logical rules. We use $90\%$ and $10\%$ of *T1 facts* for training and validation, respectively; *T2 facts* for testing. We employ two sets of labels to make our models less sensitive to the prompt format; at training time, one format is chosen with $50\%$ chance for each batch; details in Appendix F.2.

**Models.** To train larger language models, we choose the LLaMa-2 (Touvron et al., 2023) family of decoder-only models, widely adopted in literature for its performance across a variety of tasks and domains. We consider three baselines: the available pre-trained 7b and 70b models, 4-bit NormalFloat quantized (Dettmers et al., 2023), with greedy sampling strategy, temperature $t = 1.0$ and dropout disabled; we also perform supervised fine-tuning of the 7b model (4-bit, with LoRA (Hu et al., 2021)) on the ground truth T1+T2 facts set, namely "LLaMa-2-7b + XENT". We derive our LOCO-LMS fine-tuning with our proposed method LLaMa-2 7b, with 4-bit quantization and LoRA. We limit the generation to 4 tokens following the input. We adopt a similar set of hyperparameters to LoRA: we fine-tune our models for 5 epochs keeping the learning rate fixed to $\gamma = 3 \cdot 10^{-4}$, batch size 64, on 1 nVidia A100-40GB GPU. We use AdamW (Loshchilov & Hutter, 2016) as optimizer with a default weight decay $\lambda = 10^{-2}$. We use the SL to finetune three LOCO-LM variants: for negation (NEG), factual implication consistency (F-IMP) and their conjunction, i.e., given an implication $f_1 \to f_2$ we provide the SL with the constraint:

$$(z_{f_1} \oplus z_{\widetilde{f_1}}) \wedge (z_{f_1} = z_{f_1}^*) \wedge (z_{f_1} \to z_{f_2}) \wedge (z_{f_2} \oplus z_{\widetilde{f_2}}) \tag{SUPER}$$

where $\widetilde{f_1}$ and $\widetilde{f_2}$ encode the textual negation of $f_1$ and $f_2$, generated via ConCoRD's templates. We compare against orthogonal baselines such as chain-of-thought (CoT) and zero- and few-show prompting, which we note can be combined to LOCO-LMS.

**Metrics.** We fine-tune on NEG, F-IMP or SUPER and evaluate on all constraints. Specifically, we measure the implication self-consistency, cf. Equation (4), as well as the ***implication consistency***:

$$1 - |\{\alpha_i = (z_j \to z_k) : z_j^* = \top, z_k = \bot\}| \, / \, |\{\alpha_i = (z_j \to z_k) : z_j^* = \top\}| \tag{5}$$

where $z_j^*$ is the ground truth value of a fact. We also measure ***reverse implication consistency***

$$1 - |\{\alpha_i = (z_{\widetilde{k}} \to z_{\widetilde{j}}) : \neg z_k^* = \top, z_{\widetilde{j}} = \top\}| \, / \, |\{\alpha_i = (z_{\widetilde{k}} \to z_{\widetilde{j}}) : \neg z_k^* = \top\}| \tag{6}$$

and the ***reverse implication self-consistency*** variant:

$$1 - |\{\alpha_i = (z_{\widetilde{k}} \to z_{\widetilde{j}}) : z_{\widetilde{k}} = \bot, z_{\widetilde{j}} = \top\}| \, / \, |\{\alpha_i = (z_{\widetilde{k}} \to z_{\widetilde{j}}) : z_{\widetilde{k}} = \bot\}| \tag{7}$$

where $z_{\widetilde{k}}$ and $z_{\widetilde{j}}$ are the truth values of the textual negations of facts $k$ and $j$ according to the model. For negation self-consistency we compute

$$1 - |\{\alpha_i = (z_j \oplus z_{\widetilde{j}}) : z_j = z_{\widetilde{j}}\}| \, / \, |\alpha_i = (z_j \oplus z_{\widetilde{j}})|. \tag{8}$$

As in Section 5.1, we measure factuality (FAC) as the $F_1$ score on a set of ground truth facts. Finally, we account for possible shifts in the language modeling distribution by computing its perplexity (PPL) on WikiText (Merity et al., 2016), formatted as a single token sequence.

**Results.** In Table 2, we first observe an overall boost in factuality for all LOCO-LMS over the 7b baselines. Compatibly with Table 1, supervised fine-tuning is not sufficient to improve logical consistency significantly and outperforms baselines such as CoT and few-shot prompting. Our LOCO-LM trained exclusively on IMP constraints performs best in factuality and implication consistency; at the same time, scores on negation consistency and reverse implication are lower. We remark this is expected and common when doing multi-objective optimization. Note that, however, the great majority are cases of positive transfer, i.e., optimizing for one constraint also benefits others. For example, optimizing for NEG improves all columns of Table 2 wrt the baseline (C-FAC: +19%, C-IMP: +20%, C-REV: +42%, SC-REV: +35%) but self-consistency IMP, and optimizing F-IMP only degrades self-consistency REV and NEG (C-FAC: +74%, C-REV: +8%), as it rightly does

Table 2: **LOCO-LMs achieve higher (self-)consistency than off-the-shelf baselines and models trained with supervised fine-tuning** (+XENT) on the BeliefBank test split. Scores are averaged across four sets of prompts and truth labels, for which results are reported in Tables 13 and 18.

| MODEL | TRAIN | PPL | CONSISTENCY | | | SELF-CONSISTENCY | | | |
| | | | FAC | IMP | REV | NEG | IMP | REV | AVG |
|---|---|---|---|---|---|---|---|---|---|
| LLAMA-2-7B ZERO SHOT | | 52.30 | 0.27 | 0.45 | 0.47 | 0.34 | 0.45 | 0.48 | 0.41 |
| LLAMA-2-7B FEW SHOT | | 52.30 | 0.53 | 0.70 | 0.40 | 0.35 | 0.47 | 0.39 | 0.48 |
| LLAMA-2-7B COT | | 52.30 | 0.52 | 0.64 | 0.67 | 0.40 | 0.64 | 0.67 | 0.59 |
| LLAMA-2-70B ZERO SHOT | | 44.90 | 0.47 | 0.69 | 0.81 | 0.13 | 0.31 | 0.91 | 0.55 |
| LLAMA-2-7B + XENT | T1+T2 | 116.85 | 0.21 | 0.42 | 0.30 | 0.10 | 0.76 | 0.30 | 0.35 |
| LOCO-LLAMA-2-7B (NEG) | T1 | 62.21 | 0.22 | 0.50 | 0.72 | 0.48 | 0.14 | 0.68 | 0.46 |
| LOCO-LLAMA-2-7B (F-IMP) | T1 | 67.15 | **0.98** | **0.98** | 0.52 | 0.01 | **0.99** | 0.52 | 0.66 |
| LOCO-LLAMA-2-7B (SUPER) | T1 | 62.23 | 0.75 | 0.79 | **0.84** | **0.82** | 0.76 | **0.82** | **0.80** |
| LLAMA-3.1-8B ZERO SHOT | | 78.22 | 0.45 | 0.58 | 0.54 | 0.42 | 0.54 | 0.54 | 0.52 |
| LLAMA-3.1-8B FEW SHOT | | 78.22 | 0.41 | 0.54 | 0.51 | 0.36 | 0.45 | 0.51 | 0.47 |
| LOCO-LLAMA-3.1-8B (SUPER) | T1 | 78.22 | **0.73** | **0.80** | **0.80** | **0.80** | 0.76 | **0.80** | **0.78** |

Table 3: **LOCO-LMs improve Logical consistency in class-knowledge transfer** as measured on ConceptNet when trained on high-level class properties for 10 epochs.

| MODEL | CONSISTENCY | | | SELF-CONSISTENCY | | | |
| | FAC | IMP | REV | NEG | IMP | REV | AVG |
|---|---|---|---|---|---|---|---|
| LLAMA-2-7B-ZERO SHOT | 0.21 | 0.41 | **0.84** | 0.26 | 0.63 | **0.84** | 0.53 |
| LLAMA-2-7B-FEW SHOT | 0.47 | 0.69 | 0.20 | 0.10 | 0.31 | 0.20 | 0.33 |
| LOCO-LLAMA-2-7B (SUPER) | **0.72** | **0.80** | 0.59 | **0.42** | **0.80** | 0.59 | **0.65** |

not consider negation, while delivering much better performance over all cases than using XENT. Finally, ***fine-tuning a moderately-sized* LOCO-LM *on the combination of both constraints (SUPER), yields on average the most consistent language model, which on average surpasses even Llama 2 70B, a much larger model***. Overall, fine-tuning with our method does not impact negatively fluency, as measured by perplexity.

## 5.3   RQ2: CLASS KNOWLEDGE-TRANSFER IN CONCEPTNET

**Setting** We further investigate how our fine-tuning method affects the internal knowledge of an LLM by querying specific properties across hierarchies of entities. For this purpose, the ConceptNet dataset (Speer et al., 2018b), is a rich source of knowledge about entity properties and relationships. We thus construct a train split by selecting 6 high level entities (`[human, dog, cat, mammal, car, boat]` and properties of type `[CapableOf, AtLocation, IsA]`, spawning 1.227 constraints with the format e.g. `(dog, IsA, mammal)`→`(dog, CapableOf, mother of a puppy)`; similarly to experiments with BeliefBank, we fine-tune LLaMa 2 7b with our objective on the conjunction of all the considered logical constraints (SUPER) with the same hyperparameters.

**Metrics.** We construct a test set with 432 sub-entities deriving from the 6 entities considered in the train set: we consider only ground truth facts that are shared with the parent class; the underling assumption is that the LM knows the relationship `(sub-class, IsA, class)` and thus some properties should be inherited. We thus look for gains in class-knowledge transfer by comparing LOCO-LMs with the pre-trained baseline in logical consistency on sub-entity properties, which are sparse and scarce in the LM distribution. To tackle class imbalance, we augment the training set with properties that entities *don't have*, e.g. ¬`(human, CapableOf, live underwater)`.

**Results.** Table 3 indicates consistent gains from our fine-tuning method in factuality, implication (self and objective) consistency and negation self-consistency; on average, LOCO-LMs surpass base models with zero or 2 examples of factuality queries, e.g. `"Fact:  the earth is round. Label:  true"`. ***Increased factuality in* LOCO-LMs *directly reflects in implication consistency, suggesting antecedent facts learned about one class are transferred to the subordinate.***

### 5.4 RQ3: Can finetuning LoCo-LMs help consistency on unseen KB?

**Data.** We evaluate LoCo-LMs on the EntailmentBank (Dalvi et al., 2022) test split, as proposed by Kassner et al. (2023) to reason on entailment trees. It consists of 302 implication trees spawning 805 constraints, with an average of 6.57 statement nodes and 2.66 constraints per tree; we consider each node of each tree as a statement with natural language with truth label set to 1. We limit the tree depth to 5. An illustrated example is provided in Appendix 2. As in 5.2, we test two prompt and label formats. We assume that a possible semantic overlap between the training and test distributions, BeliefBank and EntailmentBank respectively, could underlie higher consistency scores across entailment trees; we estimate such overlap in Appendix E.2. Note that constraints in EntailmentBank involve more than one implication step and are akin to multi-hop reasoning.

**Competitors and Metrics.** We test our LoCo-LMs based on LLaMa-2 7b and previously trained in 5.2 on BeliefBank, without applying any changes. As baseline model, we consider LLaMa-2 7b without quantization. This experimental setup is inspired by Kassner et al. (2023), from whom we derive the notion of self-consistency on trees of entailments: each entailment tree $t \in \mathcal{T}$ is a direct acyclic graph with a single root encoding the hypothesis to be proved; a subtree $t'$ consists in each parents-child relationship in $t$, representing an entailment between the parent nodes (antecedents in logical conjunction) and the child (consequent). See Figure 2 in the Appendix for an example. For each tree $t$, we count the amount of violated subtrees $t'$, that is when a true conjunction of antecedents does not imply a true consequent. Finally, we measure logical consistency as the fraction of the total violated subtrees over the total number of subtrees in $\mathcal{T}$.

**Results.** In Table B.3 we report logical consistency across several depths. Scores are averaged across two sets of prompts and labels, detailed results are reported in Appendix B.2. We observe the consistency decreases across depths for the baseline model, until it flattens out, as more implications are evaluated. Conversely, LoCo-LM (F-IMP) and LoCo-LM (Super) achieve higher consistency across depths, validating the usefulness of our approach. *Fine-tuning LoCo-LMs on a set of constraints allows to generalize over unseen constraints* of the same type. As expected, fine-tuning for negation does not bring any added benefit (and can worsen performance) as the in EntalimentBank only implications are considered. Therefore, *our recommendation for practitioners is to fine-tune for the constraints that are considered in the downstream task, and when in doubt use a conjunction of all constraints as in our* LoCo-LMs Super which still improves w.r.t. a vanilla LLM when chaining more than two implications together.

## 6 Discussion and Further Work

Our results show that LoCo-LMs have improved (self-)consistency compared to recently introduced consistency layers which rely on external solvers, such as ConCoRD or maieutic prompting. This is *especially important for small and medium-sized LLMs*, that suffer from (self-)inconsistency and for which prompting techniques are not the final panacea (see our experiments in Section 5). In future work, we plan to extend our analysis to more complex logical operators (Vergari et al., 2021) and to consider more advanced probabilistic reasoning techniques that sport improved consistency guarantees (Ahmed et al., 2022a). Another promising direction we have not explored is that of first materializing the beliefs of an LLM such as in REFLEX (Kassner et al., 2023) and variants (Akyürek et al., 2024) and use the SL to improve consistency while potentially storing and re-using derived rules in a writable external KB (Modarressi et al., 2023; 2024).

One limitation of our approach is relying on finetuning, and thus implying sensitivity to the choice of prompt format (White et al., 2023). This can be partially addressed by fine-tuning using a mixture of formats, as we do in Section 5. While our SL is constraint-agnostic, in practice we fine-tune LoCo-LMs only against a combination of constraints known in advance. LoCo-LMs fine-tuning relies on two assumptions: that the probabilities of facts are conditionally independent given the LLM's inner state, and that the constraints in the KB are correct. The former readily applies to many LLMs, but assuming independence can bias the solutions learned by the SL (van Krieken et al., 2024). For the latter, most KBs are well-curated, but fine-tuning models against incorrect or inconsistent rules can compromise consistency and fluency.

ETHICS STATEMENT

All authors have read and approved the ICLR Code of Ethics. Concerning societal consequences, the aim of this work is to encourage LLMs to be more factual, (self-)consistent and, ultimately, reliable. However, it can also potentially enable malicious users to intentionally train LoCo-LMs against invalid rules to steer the model towards conclusions of their choice or potential reasoning shortcuts (Marconato et al., 2024b;a; Bortolotti et al., 2024). This issue is common to all strategies for aligning LLMs toward externally specified goals, like RLHF (Ouyang et al., 2022).

REPRODUCIBILITY STATEMENT

The data preprocessing pipeline is described in detail in Appendix B. Code and data released at: https://github.com/ddidacus/loco-llm.

ACKNOWLEDGMENTS

Funded by the European Union. Views and opinions expressed are however those of the author(s) only and do not necessarily reflect those of the European Union or the European Health and Digital Executive Agency (HaDEA). Neither the European Union nor the granting authority can be held responsible for them. Grant Agreement no. 101120763 - TANGO. AV is supported by the "UN-REAL: Unified Reasoning Layer for Trustworthy ML" project (EP/Y023838/1) selected by the ERC and funded by UKRI EPSRC.

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

## A    BACKGROUND ON CIRCUITS, COMPILATION AND WMC

In this section, we provide additional background and report classical results from the circuit literature (Choi et al., 2020; Vergari et al., 2021).

**Circuits** (Vergari et al., 2019; Choi et al., 2020) are constrained computational graphs that enable tractable computations. For our purposes, they enable the tractable computation of the Weighted Model Count (WMC) encoded in the semantic loss (Equation (SL)).

**Definition A.1** (Circuit). A *circuit* $c$ is a parameterized directed acyclic computational graph over variables $\mathbf{Z}$ encoding a function $c(\mathbf{Z})$, and comprising three kinds of computational units: *input*, *product*, and *sum* units. Each product or sum unit $n$ receives the outputs of other units as inputs, denoted with the set $\mathsf{in}(n)$. Each unit $n$ encodes a function $c_n$ defined as: (i) $f_n(\mathsf{sc}(n))$ if $n$ is an input unit, where $f_n$ is a function over variables $\mathsf{sc}(n) \subseteq \mathbf{Z}$, called its *scope*, (ii) $\prod_{j \in \mathsf{in}(n)} c_j(\mathsf{sc}(j))$ if $n$ is a product unit, and (iii) $\sum_{j \in \mathsf{in}(n)} w_j c_j(\mathsf{sc}(j))$ if $n$ is a sum unit, with $w_j \in \mathbb{R}$ denoting the weighted sum parameters. The scope of a product or sum unit $n$ is the union of the scopes of its inputs, i.e., $\mathsf{sc}(n) = \bigcup_{j \in \mathsf{in}(n)} \mathsf{sc}(j)$.

Tractable WMC can be achieved by ensuring that these computational graphs abide certain structural properties: *smoothness*, *decomposability* and *determinism* (Vergari et al., 2021).

**Definition A.2** (Smoothness & Decomposability). A circuit is *smooth* if for every sum unit $n$, its inputs depend on the same variables: $\forall c_1, c_2 \in \mathsf{in}(n), \mathsf{sc}(c_1) = \mathsf{sc}(c_2)$. It is *decomposable* if the inputs of every product unit $n$ depend on disjoint sets of variables: $\mathsf{in}(n) = \{c_1, c_2\}, \mathsf{sc}(c_1) \cap \mathsf{sc}(c_2) = \emptyset$.

The next step is to translate a logical constraint $\alpha_i$ into a smooth and decomposable circuit $c(\mathbf{z})$. To this end, we employ a special type of PCs, defined as follows.

**Definition A.3** (Constraint circuits). A PC $c$ over variables $\mathbf{Z}$ is a constraint circuit encoding prior knowledge $\alpha_i$ if it computes $\mathbb{1}\{\mathbf{z} \models \alpha_i\}$ for every configuration $\mathbf{z}$.

As a practical way to realize such a circuit, we will consider constraint circuits that have all sum unit parameters equal to 1 and input functionals that are indicator functions over their scope. Furthermore, we require each sum unit in it to be *deterministic*.

**Definition A.4** (Determinism). A sum unit $n$ is *deterministic* if its inputs have disjoint supports, i.e., $\forall c_1, c_2 \in \mathsf{in}(n), c_1 \neq c_2 \implies \mathsf{supp}(c_1) \cap \mathsf{supp}(c_2) = \emptyset$.

**Compilation.** We use standard compilation tools from the knowledge compilation community to turn a logical constraint into a smooth, decomposable and deterministic circuit. Specifically, we use PySDD[5] (pys, 2017) a python SDD compiler (Darwiche, 2011; Choi & Darwiche, 2013). Note that SDDs are just smooth, decomposable and deterministic circuits (Vergari et al., 2019).

Consider the following facts:

$$f_1 :\text{"an albatross is a bird"}$$
$$f_2 :\text{"an albatross breathes"}$$
$$f_3 :\text{"an albatross is an animal"}$$

and their corresponding truth values represented as three binary variables $z_1, z_2, z_3$. We want to represent the following constraint

$$(z_2 \implies z_3) \wedge (z_1 \implies z_3). \tag{9}$$

We will now sketch how a circuit compiler would proceed: the objective of compilation is to encode the above logical constraint into a *compact* form representing all possible assignments to $z_1, z_2, z_3$. We refer the reader to Choi & Darwiche (2013) for details. Our compiler proceeds in a bottom up fashion, where bottom-up compilation can be seen as composing Boolean sub-functions whose domain is determined by a variable ordering (Darwiche, 2011; Choi & Darwiche, 2013). It would start by compiling a constraint circuit that is a function of $z_1$ and $z_2$, and compose it with a constraint circuit that is a function of $z_3$ We first introduce input functionals representing indicators associated

---

[5]https://github.com/wannesm/PySDD

with each fact truth value. We will denote by $z_i$ the indicator $\mathbb{1}\{z_i = 1\}$ and by $\neg z_i$ the indicator $\mathbb{1}\{z_i = 0\}$.

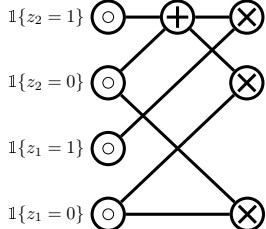

We start by disjoining the indicator $z_1$ with $\neg z_1$. This corresponds to introducing deterministic and smooth sum units in our circuits.

$\mathbb{1}\{z_2 = 1\}$ ⊙—⊕
$\mathbb{1}\{z_2 = 0\}$ ⊙

Deterministic sum units represent *disjoint solutions* to the logical formula, meaning there exists distinct assignments, characterized by the children, that satisfy the logical constraint.

The compilation process proceeds by conjoining the constraint circuits for $z_2 \vee \neg z_2$ with $z_1$, $z_2 \vee \neg z_2$ with $\neg z_1$, and $\neg z_2$ with $\neg z_1$.

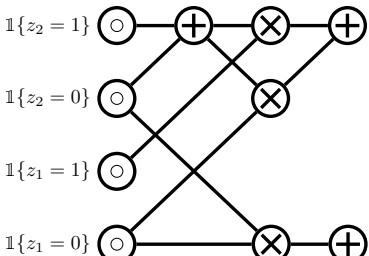

A decomposable product unit *decomposes* functions over disjoint sets of variables. The above products represent the Boolean functions $(z_2 \vee \neg z_2) \wedge z_1$, $(z_2 \vee \neg z_2) \wedge \neg z_1$, and $\neg z_1 \wedge \neg z_2$.

We disjoin $(z_2 \vee \neg z_2) \wedge z_1$ with $(z_2 \vee \neg z_2) \wedge \neg z_1$, and $\neg z_1 \wedge \neg z_2$ with true, the logical multiplicative identity.

$\mathbb{1}\{z_2 = 1\}$ ⊙—⊕—⊗—⊕
$\mathbb{1}\{z_2 = 0\}$ ⊙      ⊗
$\mathbb{1}\{z_1 = 1\}$ ⊙
$\mathbb{1}\{z_1 = 0\}$ ⊙      ⊗—⊕

So far, we have compiled constraint circuits for the logical formulas

$$((z_2 \vee \neg z_2) \wedge z_1) \vee ((z_2 \vee \neg z_2) \wedge \neg z_1)) \qquad \text{and} \qquad \neg z_1 \wedge \neg z_2.$$

We are left to conjoin the first one with $z_3$, and the second one with $\neg z_3$, and disjoin the resulting constraint circuits. What we get is a mixture over the possible solutions: If the model says that $f_1$, $f_2$, or both, are true, then it better predict that $f_3$ is true as well.

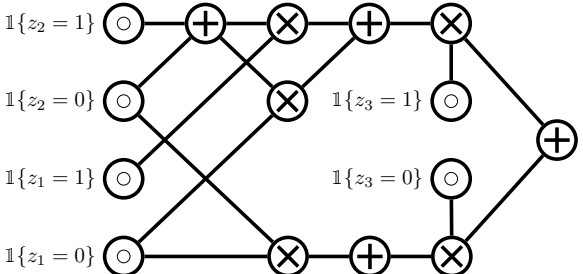

For our constraints compilation is extremely fast: taking only 2.5 milliseconds to compile a constraint and compute the loss on BeliefBank. The loss computation requires computing the WMC (Equation (SL)) in closed form. This can be done easily after compiling the logical constraint into a circuit as illustrated above. Its complexity is linear in the size of the circuit (Darwiche, 2003; Vergari et al., 2021).

**Impact on LOCO-LMS.** Thank to circuits, we can evaluate loss (Equation (SL)) *exactly* without having to explicitly enumerate truth assignments; this operation takes time linear in the size fo the circuit, yielding efficient fine-tuning.

The compilation step is also extremely fast, taking only 2.5 milliseconds to compile a constraint and compute the loss on BeliefBank. Moreover, many data points will share the same constraint during training, enabling caching.

Given ours is a pure fine-tuning approach, it has no inference-time overhead. For reference, ConCord takes 3669 seconds to perform inference on BeliefBank (silver + calibration sets) for Macaw-large, whereas LoCo applied to the same model requires only 2405 seconds.

## B    DETAILED SETTING AND RESULTS

### B.1    RQ1

#### B.1.1    DATA PREPROCESSING

We train LOCO-LMS on the BeliefBank (Kassner et al., 2021), calibration split. This dataset is derived from ConceptNet (Speer et al., 2018a), a large curated knowledge graph encoding factual knowledge and logical relations between entities at different levels of abstraction; we use the splits introduced by Mitchell et al. (Mitchell et al., 2022) for direct comparison. It consists of three pieces: a "calibration" set of $1,072$ annotated facts about 7 entities of the form *(subject, property, true/false)* used for training, a "silver" set of $12,636$ facts about 85 entities used for evaluation, and a set of 2224 valid abstract logical implications. To use our SL, we require defining a set of ground constraints. We derive these as follows. For each general implication constraint, we lookup the subjects of all facts in the training set: if the antecedent or the consequent fact of the general constraint is known for that subject, we add the subject ground constraint to the dataset $\mathcal{D}_C$.

We generate two splits: *T1 facts*, appearing either as antecedents or consequents in the constraints; *T2 facts*, appearing exclusively as consequents. The goal is to correctly guess the consequents by seeing only the antecedents and the constraints. In the calibration set, we count 796 antecedents and 276 consequents, spawning $14,005$ grounded constraints. In the silver set, we count $9,504$ antecedents and $3,132$ consequents, spawning $169,913$ grounded constraints. We subsequently test the effects of pure supervised fine-tuning: a portion of random facts from the calibration set (T1+T2) is taken with the goal to predict the excluded antecedent or consequent facts. We train on T1 facts and evaluate on T2 facts for RQ2 as well: *T1 facts* (antecedents) constitute a valid subset for all the considered logical rules.

Table 4: **LoCo-LMs achieve better logical self-consistency and factuality** as measured via Equation (4) and $F_1$ scores when compared to cross-entropy fine-tuning (XENT) and baselines using external reasoners such as ConCoRD (Mitchell et al., 2022) measured on train (calibration set) facts. For RQ1 (Section 5), LoCo-LMs fine-tuned on T1 facts only outperform training-free baseline for all metrics. For RQ2, they boost performance in the presence of a small fraction of T1+T2 facts (5-10%). For larger dataset sizes, LoCo-LMs are competitive for consistency and factuality on consequents.

|  | Method | Train size | Antecedents $F_1$ | Consequents $F_1$ | Total $F_1$ | Logical consistency |
|---|---|---|---|---|---|---|
| RQ1 | ConCoRD | | | | 0.91 | 0.91 |
| | MACAW | | 0.47 | 0.84 | 0.78 | 0.82 |
| | MACAW+XENT | T1 | 0.46 | 0.08 | 0.14 | 0.79 |
| | LoCo-LM | T1 | **0.98** | **0.99** | **0.99** | **1.00** |
| RQ2 | MACAW+XENT | T1+T2 (5%) | 0.31 | 0.73 | 0.69 | 0.90 |
| | LoCo-LM | T1+T2 (5%) | **0.34** | **0.77** | **0.72** | **0.92** |
| | MACAW+XENT | T1+T2 (10%) | 0.48 | 0.88 | 0.85 | 0.87 |
| | LoCo-LM | T1+T2 (10%) | **0.52** | **0.95** | **0.89** | **0.91** |
| | MACAW+XENT | T1+T2 (75%) | **0.69** | **1.00** | **0.97** | 0.97 |
| | LoCo-LM | T1+T2 (75%) | 0.65 | **1.00** | **0.97** | **0.99** |

## B.2 RQ2

Table 5: **LoCo-LMs evaluated on BeliefBank, training (calibration) split.** Scores are averaged across four prompt formats and truth labels. We observe fine-tuning with our method allows for higher logical consistency to different rules.

| MODEL | TRAIN SUBSET | PPL | CONSISTENCY | | | SELF-CONSISTENCY | | | AVG |
|---|---|---|---|---|---|---|---|---|---|
| | | | FAC | IMP | REV | NEG | IMP | REV | |
| LLaMa-2-7b Zero Shot | | 52.30 | 0.29 | 0.47 | 0.48 | 0.33 | 0.44 | 0.50 | 0.42 |
| LLaMa-2-7b Few Shot | | 52.30 | 0.55 | 0.72 | 0.42 | 0.36 | 0.47 | 0.42 | 0.49 |
| LLaMa-2-7b + XENT | T1+T2 | 116.85 | 0.14 | 0.35 | 0.47 | 0.11 | 0.57 | 0.47 | 0.31 |
| LoCo-LLaMa-2-7b (NEG) | T1 | 62.21 | 0.14 | 0.41 | 0.71 | 0.41 | 0.28 | 0.68 | 0.44 |
| LoCo-LLaMa-2-7b (F-IMP) | T1 | 67.15 | **1.00** | **1.00** | 0.52 | 0.00 | **1.00** | 0.52 | 0.68 |
| LoCo-LLaMa-2-7b (SUPER) | T1 | 62.23 | 0.86 | 0.91 | **0.81** | **0.85** | 0.84 | **0.82** | **0.85** |
| LLaMa-3.1-8b Zero Shot | | 78.22 | 0.44 | 0.55 | 0.57 | 0.38 | 0.54 | 0.57 | 0.52 |
| LLaMa-3.1-8b Few Shot | | 78.22 | 0.41 | 0.53 | 0.48 | 0.36 | 0.45 | 0.48 | 0.44 |
| LoCo-LLaMa-3.1-8b (SUPER) | T1 | 78.22 | **0.82** | **0.89** | **0.84** | **0.81** | **0.81** | **0.84** | **0.83** |

Table 6: **LoCo-LMs evaluated on BeliefBank, training (calibration) split.** Prompt format 1 `[true, false]` is used. We observe fine-tuning with our method allows for higher logical consistency to different rules.

| MODEL | TRAIN SUBSET | PPL | CONSISTENCY | | | SELF-CONSISTENCY | | | |
| | | | FAC | IMP | REV | NEG | IMP | REV | AVG |
|---|---|---|---|---|---|---|---|---|---|
| LLAMA-2-7B ZERO SHOT | | 52.30 | 0.43 | 0.63 | 0.33 | 0.38 | 0.29 | 0.39 | 0.41 |
| LLAMA-2-7B FEW SHOT | | 52.30 | 0.53 | 0.74 | 0.36 | 0.28 | 0.42 | 0.37 | 0.45 |
| LLAMA-2-7B COT | | 52.30 | 0.67 | 0.76 | 0.77 | 0.32 | 0.74 | 0.77 | 0.66 |
| LLAMA-2-70B ZERO SHOT | | 44.90 | 0.52 | 0,76 | 0.79 | 0.18 | 0.35 | 0.90 | 0.58 |
| LLAMA-2-7B + XENT | T1+T2 | 116.85 | 0.37 | 0.47 | 0.02 | 0.16 | 0.89 | 0.02 | 0.32 |
| LOCO-LLAMA-2-7B (NEG) | T1 | 62.21 | 0.46 | 0.70 | **0.85** | 0.93 | 0.28 | 0.72 | 0.66 |
| LOCO-LLAMA-2-7B (F-IMP) | T1 | 67.15 | **1.00** | **1.00** | 0.08 | 0.00 | **1.00** | 0.08 | 0.53 |
| LOCO-LLAMA-2-7B (SUPER) | T1 | 62.23 | 0.88 | 0.91 | 0.72 | **0.94** | 0.86 | **0.73** | **0.84** |
| LLAMA-3.1-8B ZERO SHOT | | 78.22 | 0.47 | 0.58 | 0.63 | 0.48 | 0.61 | 0.63 | 0.57 |
| LLAMA-3.1-8B FEW SHOT | | 78.22 | 0.45 | 0.55 | 0.57 | 0.47 | 0.52 | 0.57 | 0.52 |
| LLAMA-3.1-8B (SUPER) | T1 | 78.22 | **0.80** | **0.89** | **0.79** | **0.76** | **0.81** | **0.79** | **0.81** |

Table 7: **LoCo-LMs evaluated on BeliefBank, training (calibration) split.** Prompt format 2 `[yes, no]` is used. We observe fine-tuning with our method allows for higher logical consistency to different rules.

| MODEL | TRAIN SUBSET | PPL | CONSISTENCY | | | SELF-CONSISTENCY | | | |
| | | | FAC | IMP | REV | NEG | IMP | REV | AVG |
|---|---|---|---|---|---|---|---|---|---|
| LLAMA-2-7B ZERO SHOT | | 52.30 | 0.39 | 0.51 | 0.08 | 0.46 | 0.27 | 0.09 | 0.30 |
| LLAMA-2-7B FEW SHOT | | 52.30 | 0.52 | 0.66 | 0.55 | 0.48 | 0.55 | 0.55 | 0.55 |
| LLAMA-2-7B COT | | 52.30 | 0.38 | 0.52 | 0.57 | 0.48 | 0.54 | 0.57 | 0.51 |
| LLAMA-2-70B ZERO SHOT | | 44.90 | 0.46 | 0.68 | 0.81 | 0.05 | 0.28 | 0.93 | 0.54 |
| LLAMA-2-7B + XENT | T1+T2 | 116.85 | 0.05 | 0.32 | 0.00 | 0.04 | 0.00 | 0.00 | 0.07 |
| LOCO-LLAMA-2-7B (NEG) | T1 | 62.21 | 0.09 | 0.33 | 0.00 | 0.70 | 0.82 | 0.00 | 0.32 |
| LOCO-LLAMA-2-7B (F-IMP) | T1 | 67.15 | **1.00** | **1.00** | 0.08 | 0.00 | **1.00** | 0.08 | 0.53 |
| LOCO-LLAMA-2-7B (SUPER) | T1 | 62.23 | 0.84 | 0.87 | **0.79** | 0.82 | 0.74 | **0.80** | **0.81** |
| LLAMA-3.1-8B ZERO SHOT | | 78.22 | 0.43 | 0.49 | 0.75 | 0.44 | 0.57 | 0.75 | 0.57 |
| LLAMA-3.1-8B FEW SHOT | | 78.22 | 0.31 | 0.42 | 0.51 | 0.31 | 0.42 | 0.51 | 0.43 |
| LLAMA-3.1-8B (SUPER) | | 78.22 | **0.81** | **0.89** | **0.84** | **0.78** | **0.78** | **0.84** | **0.82** |

Table 8: **LoCo-LMs evaluated on BeliefBank, training (calibration) split.** Prompt format 3 `[correct, incorrect]` is used. We observe fine-tuning with our method allows for higher logical consistency to different rules.

| MODEL | TRAIN SUBSET | PPL | CONSISTENCY | | | SELF-CONSISTENCY | | | |
| | | | FAC | IMP | REV | NEG | IMP | REV | AVG |
|---|---|---|---|---|---|---|---|---|---|
| LLAMA-2-7B ZERO SHOT | | 52.30 | 0.29 | 0.42 | 0.55 | 0.44 | 0.51 | 0.55 | 0.46 |
| LLAMA-2-7B FEW SHOT | | 52.30 | 0.46 | 0.69 | 0.00 | 0.00 | 0.28 | 0.00 | 0.24 |
| LLAMA-2-7B + XENT | T1+T2 | 116.85 | 0.10 | 0.31 | 0.86 | 0.20 | 0.65 | 0.86 | 0.50 |
| LOCO-LLAMA-2-7B (NEG) | T1 | 62.21 | 0.00 | 0.30 | **1.00** | 0.02 | 0.00 | **1.00** | 0.39 |
| LOCO-LLAMA-2-7B (F-IMP) | T1 | 67.15 | **0.99** | **1.00** | 0.96 | 0.01 | 1.00 | 0.96 | 0.82 |
| LOCO-LLAMA-2-7B (SUPER) | T1 | 62.23 | 0.80 | 0.92 | 0.80 | **0.80** | **0.85** | 0.80 | **0.84** |
| LLAMA-3.1-8B ZERO SHOT | | 78.22 | 0.43 | 0.63 | 0.13 | 0.25 | 0.34 | 0.13 | 0.32 |
| LLAMA-3.1-8B FEW SHOT | | 78.22 | 0.40 | 0.56 | 0.15 | 0.19 | 0.31 | 0.15 | 0.29 |
| LOCO-LLAMA-3.1-8B (SUPER) | T1 | 78.22 | **0.79** | **0.86** | **0.83** | **0.80** | **0.78** | **0.83** | **0.82** |

## B.3 RQ3

Table 9: **LoCo-LMs evaluated on BeliefBank, training (calibration) split.** Prompt format 4 `[right, wrong]` is used. We observe fine-tuning with our method allows for higher logical consistency to different rules.

| MODEL | TRAIN SUBSET | PPL | CONSISTENCY | | | SELF-CONSISTENCY | | | |
| --- | --- | --- | --- | --- | --- | --- | --- | --- | --- |
| | | | FAC | IMP | REV | NEG | IMP | REV | AVG |
| LLaMa-2-7b Zero Shot | | 52.30 | 0.05 | 0.31 | 0.96 | 0.05 | 0.67 | 0.96 | 0.50 |
| LLaMa-2-7b Few Shot | | 52.30 | 0.69 | 0.78 | 0.76 | 0.66 | 0.64 | 0.76 | 0.71 |
| LLaMa-2-7b + XENT | T1+T2 | 116.85 | 0.02 | 0.29 | 0.98 | 0.04 | 0.75 | 0.98 | 0.34 |
| LoCo-LLaMa-2-7b (NEG) | T1 | 62.21 | 0.00 | 0.30 | **1.00** | 0.00 | 0.00 | **1.00** | 0.38 |
| LoCo-LLaMa-2-7b (F-IMP) | T1 | 67.15 | **0.99** | **1.00** | 0.96 | 0.00 | **1.00** | 0.96 | 0.82 |
| LoCo-LLaMa-2-7b (Super) | T1 | 62.23 | 0.91 | 0.92 | 0.94 | **0.83** | 0.90 | 0.94 | **0.91** |
| LLaMa-3.1-8b Zero Shot | | 78.22 | 0.43 | 0.50 | 0.75 | 0.34 | 0.63 | 0.75 | 0.62 |
| LLaMa-3.1-8b Few Shot | | 78.22 | 0.48 | 0.57 | 0.67 | 0.46 | 0.56 | 0.67 | 0.52 |
| LoCo-LLaMa-3.1-8b (Super) | T1 | 78.22 | **0.89** | **0.90** | **0.90** | **0.90** | **0.86** | **0.90** | **0.89** |

Table 10: **LoCo-LMs evaluated on BeliefBank, test (silver) split.** Prompt format 1 `[true, false]` is used. We observe fine-tuning with our method allows for higher logical consistency to different rules.

| MODEL | TRAIN SUBSET | PPL | CONSISTENCY | | | SELF-CONSISTENCY | | | |
| --- | --- | --- | --- | --- | --- | --- | --- | --- | --- |
| | | | FAC | IMP | REV | NEG | IMP | REV | AVG |
| LLaMa-2-7b Zero Shot | | 52.30 | 0.41 | 0.55 | 0.22 | 0.41 | 0.30 | 0.25 | 0.36 |
| LLaMa-2-7b Few Shot | | 52.30 | 0.53 | 0.75 | 0.37 | 0.27 | 0.41 | 0.37 | 0.45 |
| LLaMa-2-7b CoT | | 52.30 | 0.67 | 0.76 | 0.77 | 0.32 | 0.74 | 0.77 | 0.67 |
| LLaMa-2-70b Zero Shot | | 44.90 | 0.50 | 0.72 | 0.80 | 0.20 | 0.34 | **0.89** | 0.58 |
| LLaMa-2-7b + XENT | T1+T2 | 116.85 | 0.40 | 0.52 | 0.02 | 0.11 | 0.82 | 0.02 | 0.31 |
| LoCo-LLaMa-2-7b (NEG) | T1 | 62.21 | 0.44 | 0.64 | **0.86** | **0.92** | 0.28 | 0.72 | 0.64 |
| LoCo-LLaMa-2-7b (F-IMP) | T1 | 67.15 | **0.98** | **0.98** | 0.07 | 0.00 | **0.98** | 0.07 | 0.51 |
| LoCo-LLaMa-2-7b (Super) | T1 | 62.23 | 0.75 | 0.78 | 0.72 | 0.91 | 0.74 | 0.72 | **0.77** |
| LLaMa-3.1-8b Zero Shot | | 78.22 | 0.46 | 0.60 | 0.65 | 0.50 | 0.60 | 0.65 | 0.59 |
| LLaMa-3.1-8b Few Shot | | 78.22 | 0.48 | 0.60 | 0.65 | 0.49 | 0.55 | 0.65 | 0.57 |
| LoCo-LLaMa-3.1-8b (Super) | T1 | 78.22 | **0.71** | **0.81** | **0.72** | **0.71** | **0.74** | **0.72** | **0.74** |

Table 11: **LoCo-LMs evaluated on BeliefBank, test (silver) split.** Prompt format 2 `[yes, no]` is used. We observe fine-tuning with our method allows for higher logical consistency to different rules.

| MODEL | TRAIN SUBSET | PPL | CONSISTENCY | | | SELF-CONSISTENCY | | | |
| --- | --- | --- | --- | --- | --- | --- | --- | --- | --- |
| | | | FAC | IMP | REV | NEG | IMP | REV | AVG |
| LLaMa-2-7b Zero Shot | | 52.30 | 0.37 | 0.48 | 0.04 | 0.43 | 0.29 | 0.04 | 0.28 |
| LLaMa-2-7b Few Shot | | 52.30 | 0.53 | 0.67 | 0.57 | 0.49 | 0.58 | 0.53 | 0.56 |
| LLaMa-2-7b CoT | | 52.30 | 0.38 | 0.52 | 0.57 | 0.48 | 0.54 | 0.57 | 0.51 |
| LLaMa-2-70b Zero Shot | | 44.90 | 0.44 | 0.65 | **0.82** | 0.05 | 0.29 | **0.93** | 0.53 |
| LLaMa-2-7b + XENT | T1+T2 | 116.85 | 0.11 | 0.39 | 0.00 | 0.03 | 0.80 | 0.00 | 0.22 |
| LoCo-LLaMa-2-7b (NEG) | T1 | 62.21 | 0.44 | 0.65 | 0.00 | **1.00** | 0.28 | 0.00 | 0.40 |
| LoCo-LLaMa-2-7b (F-IMP) | T1 | 67.15 | **0.99** | **0.99** | 0.07 | 0.00 | **0.99** | 0.07 | 0.52 |
| LoCo-LLaMa-2-7b (Super) | T1 | 62.23 | 0.73 | 0.75 | 0.81 | 0.83 | 0.67 | 0.82 | **0.77** |
| LLaMa-3.1-8b Zero Shot | | 78.22 | 0.44 | 0.55 | 0.72 | 0.42 | 0.61 | 0.72 | 0.58 |
| LLaMa-3.1-8b Few Shot | | 78.22 | 0.33 | 0.46 | 0.54 | 0.32 | 0.42 | 0.54 | 0.43 |
| LoCo-LLaMa-3.1-8b (Super) | T1 | 78.22 | **0.71** | **0.81** | **0.75** | **0.76** | **0.74** | **0.75** | **0.75** |

# C  ConceptNet

Table 12: **LOCO-LMS evaluated on BeliefBank, test (silver) split.** Prompt format 3 `[correct, incorrect]` is used. We observe fine-tuning with our method allows for higher logical consistency to different rules.

| MODEL | TRAIN SUBSET | PPL | CONSISTENCY | | | SELF-CONSISTENCY | | | AVG |
|---|---|---|---|---|---|---|---|---|---|
| | | | FAC | IMP | REV | NEG | IMP | REV | |
| LLAMA-2-7B ZERO SHOT | | 52.30 | 0.25 | 0.40 | 0.65 | 0.45 | 0.52 | 0.65 | 0.49 |
| LLAMA-2-7B FEW SHOT | | 52.30 | 0.44 | 0.64 | 0.00 | 0.00 | 0.28 | 0.00 | 0.23 |
| LLAMA-2-7B + XENT | T1+T2 | 116.85 | 0.12 | 0.35 | 0.89 | 0.17 | 0.66 | 0.89 | 0.51 |
| LOCO-LLAMA-2-7B (NEG) | T1 | 62.21 | 0.00 | 0.35 | **1.00** | 0.01 | 0.00 | **1.00** | 0.39 |
| LOCO-LLAMA-2-7B (F-IMP) | T1 | 67.15 | **0.98** | **0.98** | 0.96 | 0.01 | **0.99** | 0.96 | **0.81** |
| LOCO-LLAMA-2-7B (SUPER) | T1 | 62.23 | 0.73 | 0.83 | 0.92 | **0.76** | 0.80 | 0.82 | **0.81** |
| LLAMA-3.1-8B ZERO SHOT | | 78.22 | 0.42 | 0.60 | 0.15 | 0.26 | 0.34 | 0.15 | 0.32 |
| LLAMA-3.1-8B FEW SHOT | | 78.22 | 0.40 | 0.55 | 0.15 | 0.19 | 0.31 | 0.15 | 0.31 |
| LOCO-LLAMA-3.1-8B (SUPER) | T1 | 78.22 | **0.72** | **0.79** | **0.83** | **0.81** | **0.77** | **0.83** | **0.79** |

Table 13: **LOCO-LMS evaluated on BeliefBank, test (silver) split.** Prompt format 4 `[right, wrong]` is used. We observe fine-tuning with our method allows for higher logical consistency to different rules.

| MODEL | TRAIN SUBSET | PPL | CONSISTENCY | | | SELF-CONSISTENCY | | | AVG |
|---|---|---|---|---|---|---|---|---|---|
| | | | FAC | IMP | REV | NEG | IMP | REV | |
| LLAMA-2-7B ZERO SHOT | | 52.30 | 0.03 | 0.35 | 0.97 | 0.05 | 0.67 | 0.97 | 0.51 |
| LLAMA-2-7B FEW SHOT | | 52.30 | 0.63 | 0.75 | 0.67 | 0.64 | 0.62 | 0.67 | 0.66 |
| LLAMA-2-7B + XENT | T1+T2 | 116.85 | 0.21 | 0.42 | 0.30 | 0.10 | 0.76 | 0.30 | 0.35 |
| LOCO-LLAMA-2-7B (NEG) | T1 | 62.21 | 0.00 | 0.35 | **1.00** | 0.00 | 0.00 | **1.00** | 0.39 |
| LOCO-LLAMA-2-7B (F-IMP) | T1 | 67.15 | **0.98** | **0.98** | 0.96 | 0.01 | **0.99** | 0.96 | 0.81 |
| LOCO-LLAMA-2-7B (SUPER) | T1 | 62.23 | 0.80 | 0.80 | 0.91 | **0.78** | 0.81 | 0.91 | **0.83** |
| LLAMA-3.1-8B ZERO SHOT | | 78.22 | 0.47 | 0.58 | 0.63 | 0.48 | 0.61 | 0.63 | 0.57 |
| LLAMA-3.1-8B FEW SHOT | | 78.22 | 0.43 | 0.53 | 0.68 | 0.44 | 0.52 | 0.68 | 0.55 |
| LOCO-LLAMA-3.1-8B (SUPER) | T1 | 78.22 | **0.77** | **0.80** | **0.89** | **0.90** | **0.80** | **0.89** | **0.84** |

Table 14: **LOCO-LMS can achieve higher consistency across depth than the baseline.** Scores are computed with Format 1 `[true, false]`, reported in Appendix F.2. LOCO-LM fine-tuned with on the implication rule achieves best consistency.

| MODEL | DEPTH | | | | |
|---|---|---|---|---|---|
| | 1 | 2 | 3 | 4 | 5 |
| LLAMA-2-7B | 0.73 | 0.77 | 0.79 | 0.80 | 0.80 |
| LOCO-LLAMA-2-7B (NEG) | 0.03 | 0.03 | 0.03 | 0.04 | 0.05 |
| LOCO-LLAMA-2-7B (F-IMP) | **0.97** | **0.96** | **0.97** | **0.97** | **0.97** |
| LOCO-LLAMA-2-7B (SUPER) | 0.75 | 0.74 | 0.73 | 0.73 | 0.74 |

# D ENTAILMENTBANK

Table 15: **LoCo-LMs can be consistent across unseen trees of entailments** when trained for implication consistency (F-IMP) on BeliefBank and evaluated as is on EntailmentBank (Dalvi et al., 2022).

| model | depth | | | | |
|---|---|---|---|---|---|
| | 1 | 2 | 3 | 4 | 5 |
| LLaMa-2-7b | 0.87 | 0.76 | 0.59 | 0.61 | 0.63 |
| LoCo-LLaMa-2-7b (NEG) | 0.51 | 0.51 | 0.51 | 0.52 | 0.52 |
| LoCo-LLaMa-2-7b (F-IMP) | **0.98** | **0.98** | **0.98** | **0.98** | **0.98** |
| LoCo-LLaMa-2-7b (Super) | 0.69 | 0.68 | 0.68 | 0.68 | 0.69 |

Table 16: **LoCo-LMs can achieve higher consistency across depth than the baseline.** Scores are computed with Format 2 [yes, no], reported in Appendix F.2. LoCo-LM fine-tuned with on the implication rule and the negation rule achieve best consistency.

| MODEL | DEPTH | | | | |
|---|---|---|---|---|---|
| | 1 | 2 | 3 | 4 | 5 |
| LLAMA-2-7B | 1.00 | 0.75 | 0.38 | 0.42 | 0.46 |
| LOCO-LLAMA-2-7B (NEG) | **0.99** | **0.99** | **0.99** | **0.99** | **0.99** |
| LOCO-LLAMA-2-7B (F-IMP) | **0.99** | **0.99** | **0.99** | **0.99** | **0.99** |
| LOCO-LLAMA-2-7B (SUPER) | 0.62 | 0.62 | 0.63 | 0.63 | 0.64 |

Table 17: Distribution of answer labels from LoCo-LMs for different prompt formats on the EntailmentBank test split.

| MODEL | LABELS: [YES, NO] | | | LABELS: [TRUE, FALSE] | | |
|---|---|---|---|---|---|---|
| | YES | NO | INVALID | TRUE | FALSE | INVALID |
| LLAMA-2-7B | 1188 | 6 | 1441 | 615 | 1742 | 278 |
| LOCO-LLAMA-2-7B (NEG) | 2538 | 0 | 97 | 940 | 0 | 1695 |
| LOCO-LLAMA-2-7B (F-IMP) | 2557 | 0 | 78 | 2441 | 194 | 0 |
| LOCO-LLAMA-2-7B (SUPER) | 2079 | 486 | 70 | 874 | 1756 | 5 |

Table 18: **LoCo-LMs evaluated on BeliefBank, train (calibration) split, compared by decoding strategy.** All few shot prompts in Appendix F.3 were used. The default configuration consists in top_k = 50, top_p = 1.0, temperature = 1.0; greedy decoding consists in top_k = 1, top_p = 1.0, temperature = 1.0. We observe no significant difference as in the current setup, the truth label is represented by a single token and thus changes in the sampling technique could be better observed on an output sequence.

| MODEL | DECODING | PPL | CONSISTENCY | | | SELF-CONSISTENCY | | | |
|---|---|---|---|---|---|---|---|---|---|
| | | | FAC | IMP | REV | NEG | IMP | REV | AVG |
| LOCO-LLAMA-2-7B (SUPER) | DEFAULT | 62.41 | 0.79 | 0.83 | 0.82 | 0.57 | 0.76 | 0.82 | 0.76 |
| LOCO-LLAMA-2-7B (SUPER) | GREEDY | 62.41 | 0.79 | 0.82 | 0.82 | 0.57 | 0.76 | 0.82 | 0.76 |

# E    SEMANTIC OVERLAP

We base our measurement for semantic overlap on cosine similarity, widely adopted in literature. We report our results with a note for caution: it is unclear whether embeddings could be similar for the semantic features we are seeking Steck et al. (2024), suggesting further research on the topic.

Table 19: Logical consistency in class-knowledge transfer measured in LoCo-LMs on ConceptNet with Format 2 [yes, no]. On average, LoCo-LMs trained on joint logical constraints surpass baseline methods, with consistent gains in factuality and implication consistency. LoCo-LMs have been trained on high-level class properties for 10 epochs.

| | CONSISTENCY | | | SELF-CONSISTENCY | | | |
| MODEL | FAC | IMP | REV | NEG | IMP | REV | AVG |
|---|---|---|---|---|---|---|---|
| LLaMa-2-7b-zero shot | 0.24 | 0.41 | **0.83** | 0.26 | 0.63 | **0.83** | 0.53 |
| LLaMa-2-7b-few shot | 0.56 | 0.71 | 0.56 | **0.48** | 0.56 | 0.56 | 0.57 |
| LoCo-LLaMa-2-7b (Super) | **0.74** | **0.82** | 0.59 | 0.41 | **0.83** | 0.59 | **0.66** |

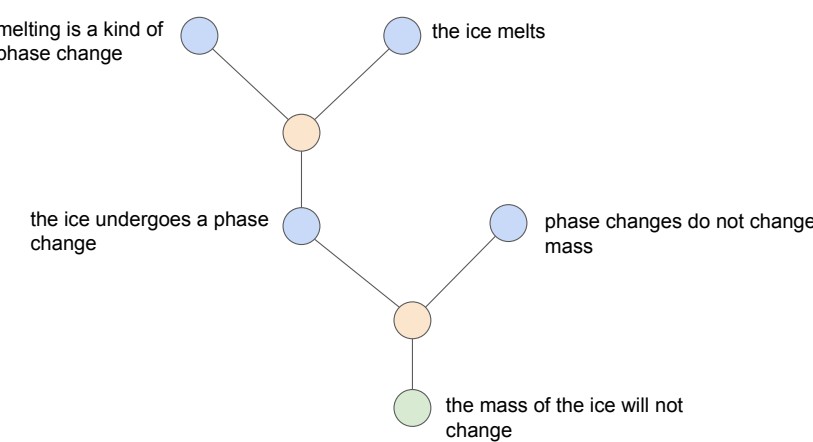

Figure 2: An illustration of an entailment tree, namely a "prof", from EntailmentBank Dalvi et al. (2022). Blue nodes are premises in logical conjunction, orange nodes are implications and the green node denote the hypothesis to prove.

## E.1 BeliefBank

We measure the semantic overlap between the training and test distribution by constructing a Representation Dissimilarity Matrix (RDM) of Macaw's embeddings (token average) between training and test entities. The main assumption is that semantically similar subjects may have similar properties, as a proxy for domain knowledge transfer.

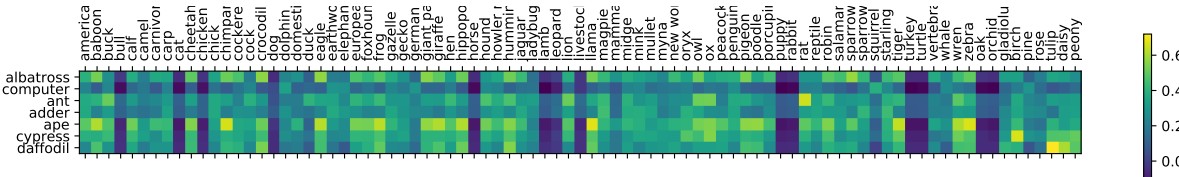

Figure 3: Pairwise cosine similarities between entities in the training distribution (calibration, rows) and the test distribution (silver, columns).

## E.2 BeliefBank-EntailmentBank

We consider the training split, namely "calibration" in ConCoRD Mitchell et al. (2022), from BeliefBank Kassner et al. (2021), and the test split from EntailmentBank Dalvi et al. (2022) to estimate

Table 20: Fraction $k$ of facts in BeliefBank with cosine similarity above $t$ with any fact in EntailmentBank, for $t = \{0.80, 0.85, 0.90\}$.

| $t$ | $k$ |
|------|------|
| 0.80 | 0.41 |
| 0.85 | 0.22 |
| 0.90 | 0.02 |

the knowledge that LOCO-LMS could transfer to entailment trees. We process BeliefBank as a set of $1,072$ facts, while EntailmentBank as a set of $2,635$ facts. Both sets contain statements in natural language that are converted into vector embeddings through encoding with LLaMa-2-7b Touvron et al. (2023); the last layer logits are considered and a sentence representation is obtained by averaging across tokens. We consequently compute the pairwise cosine similarities between fact embeddings from both sets. For each fact in BeliefBank, we take the maximum similarity with any fact from EntailmentBank, which should represent the existance of a unit of a similar knowledge between the two datasets. Given the volume of pairwise comparisons, we aggregate the results.

## F PROMPTS

### F.1 PROMPTS FOR MACAW-LARGE

We query the language model for a belief label about a statement in natural language. We adopt the format:

> **Prompt**
>
> $answer$ ; $mcoptions$ = (A) pos_label (B) neg_label ; $question$ = Is subject a property?

We fix `<pos_label>` = "Yes." and `<neg_label>` = "No.". We converted the (`<subject>`, `<property>`) tuple in natural language with a formatting function provided by Mitchell et al. (Mitchell et al., 2022).

> **Expected answers**
>
> $answer$ = pos_label ; $answer$ = neg_label ;

### F.2 ZERO-SHOT PROMPTS FOR LOCO-LMS

We adopt two label sets to make the model less *prompt sensitive*:

**Format 1**: `[true, false]`

> **Prompt**
>
> You can answer only with "true" or "false". Is the fact true? Fact: statement

> **Expected answers**
>
> Answer: true
> Answer: false

**Format 2**: `[yes, no]`

**Prompt**

You can answer only with "yes" or "no". Is the fact true? Fact: statement

**Expected answers**

Answer: yes
Answer: no

**Format 3**: `[correct, incorrect]`

**Prompt**

You can answer only with "correct" or "incorrect". Is the fact true? Fact: statement

**Expected answers**

Answer: correct
Answer: incorrect

**Format 4**: `[right, wrong]`

**Prompt**

You can answer only with "right" or "wrong". Is the fact true? Fact: statement

**Expected answers**

Answer: right
Answer: wrong

### F.3 FEW-SHOT PROMPTS FOR LOCO-LMS

We adopt two label sets to make the model less *prompt sensitive*:

**Format 1**: `[true, false]`

**Prompt**

Fact: the earth is round. Label: true.
Fact: the sun is cold. Label: false.
Fact: {fact}. Label:

**Expected answers**

Answer: true
Answer: false

**Format 2**: `[yes, no]`

**Prompt**

Fact: the earth is round. Label: yes.
Fact: the sun is cold. Label: no.
Fact: {fact}. Label:

**Expected answers**

Answer: yes
Answer: no

**Format 3**: `[correct, incorrect]`

**Prompt**

Statement: the earth is round. Label: yes.
Statement: the sun is cold. Label: no.
Statement: {fact}. Label:

**Expected answers**

Answer: correct
Answer: correct

**Format 4**: `[right, wrong]`

**Prompt**

Claim: the earth is round. Label: yes.
Claim: the sun is cold. Label: no.
Claim: {fact}. Label:

**Expected answers**

Answer: right
Answer: wrong

