# OpenReview forum: "Logically Consistent Language Models via Neuro-Symbolic Integration"
_ICLR.cc/2025/Conference — ICLR 2025 Poster_

### Official Review · Reviewer_G8eP · 2024-11-02

**Soundness:** 3
**Presentation:** 3
**Contribution:** 3
**Rating:** 6
**Confidence:** 3

**Summary:**

This paper introduces LOCO-LMS, a fine-tuning method grounded in neural-symbolic reasoning, which significantly enhances the logical consistency and factuality of LLMs by integrating logical constraints as loss functions during training. Unlike traditional methods that rely on external reasoning tools, LOCO-LMS internalizes logical rules, allowing the model to reason independently and improving overall efficiency.

**Strengths:**

1.  LOCO-LMS effectively improves the model's logical consistency, accommodating complex logical relationships such as positive implication, reverse implication, and negation. This alignment with common sense enhances the quality of responses generated by LLMs.

2.  By incorporating semantic loss, the method minimizes reliance on external reasoning tools, thereby lowering reasoning costs and increasing inference speed.

**Weaknesses:**

1. The model assumes that facts are conditionally independent under a given model state, but in actual applications, there may be dependencies between facts, and this assumption may affect the consistency effect.

2. While it addresses factual inconsistencies in the Llama-7B model, I also concern that its efficiency and scalability may lag behind approaches based on RAG and knowledge editing.

3. LOCO-LMS is designed for specific tasks and fine-tuning, which limits its applicability to more complex reasoning tasks. Additionally, it may be vulnerable to attacks, such as just-in-time injection.

**Questions:**

1. Can LOCO-LMS be adapted for more complex, multi-level, or nonlinear logical reasoning scenarios?

2. How well does LOCO-LMS integrate with existing knowledge editing methods when it comes to incorporating new facts or updating knowledge bases?

---

> ### Author Response · Authors · 2024-11-21
>
> We thank the reviewer for the feedback and for appreciating how our approach manages to efficiently improve consistency and quality of responses without needing an external solver. We address below all the concerns they raised.
>
> > *LoCo-LLMs assumes that facts are conditionally independent*
>
> Good point. This limitation is actually shared by most works in Neuro-Symbolic AI, including circuit-based solutions [1]. This does not have a dramatic impact on performance, however, as highlighted by our experiments.  We will consider relaxing this assumption in future work.
>
> [1] van Krieken et al. "On the Independence Assumption in Neurosymbolic Learning." Forty-first International Conference on Machine Learning. (2024)
>
> > *efficiency and scalability may lag behind approaches [like] RAG and knowledge editing.*
>
> Concerning efficiency, LoCo-LMs only require fine-tuning using a light-weight loss term and have no inference-time cost.  At training time, our loss leverages circuits to avoid having to enumerate truth assignments, and allow us to compute exact probabilities of satisfying constraints into an operation that takes time linear in the size fo the circuit. (see lines 174-177 for more refs). The compilation step is also extremely fast, taking only ~2.5 milliseconds to compile a constraint and compute the loss on BeliefBank. Moreover, many data points will share the same constraint during training, enabling caching.  At inference time, our approach has no overhead. For reference, ConCord takes ~3669 seconds to perform inference on BeliefBank (silver + calibration sets) for Macaw-large, whereas LoCo applied to the same model only requires only ~2405 seconds. We will add these results at the end of Appendix A.
>
> While RAG has been used to mitigate factuality hallucinations [Lewis et al., 2020], it is no silver bullet, as LLMs occasionally ignore retrieved information [Xu et al., 2024, Jin et al., 2024], rather over relying on their learned knowledge.  LoCo-LLMs are designed to avoid this.
>
> [Lewis et al., 2020] Lewis et al. "Retrieval-augmented generation for knowledge-intensive nlp tasks." Advances in Neural Information Processing Systems 33 (2020).
>
> [Xu et al., 2024] Xu et al. "Knowledge conflicts for llms: A survey." arXiv:2403.08319 (2024).
>
> [Jin et al., 2024] Jin et al. "Tug-of-war between knowledge: Exploring and resolving knowledge conflicts in retrieval-augmented language models." arXiv preprint arXiv:2402.14409 (2024).
>
> We believe LoCo-LMs, which are more self-consistent and factual compared to regular LLMs, could benefit knowledge editing, in the sense that 1) it makes it less likely we need to edit LLMs to achieve the same objectives, allowing us to focus on updating their knowledge instead, and 2) updating a more self-consistent model can be less likely to produce non-logical “ripple effects”. Intuitively, the “ripple effects” left could be of a “logical kind” and thus making it easier to identify and correct them using logical consistency. This is a very interesting research question to investigate as future work.
>
> > *LoCo-LLMs may be vulnerable to attacks, such as just-in-time injection.*
>
> We think LoCo-LLMs are no more vulnerable to prompt injection attacks than regular LLMs.
> Please let us know if you have further pointers that can suggest the opposite, we are happy to investigate this further.
>
> > *Can LOCO-LMS be adapted for more complex, multi-level, or nonlinear logical reasoning scenarios?*
>
> We stress that the Semantic Loss term is applicable to logic formulas with an arbitrary number of connectives and logic variables.  We showcased this property in our EntailmentBank experiment (Section 5.3), where the goal is to ensure consistency wrt *entailment trees* with up to 10+ logic facts. (The number of facts ranges from 1 to 5, see Figure 2 of [Dalvi et al., 2022] for the precise distribution.)
>
> We are not sure about the second point: could you please clarify what you mean by nonlinear logical reasoning?
>
> [Dalvi et al., 2022] Dalvi et al. Explaining answers with entailment trees. EMNLP 2022.

---

### Official Review · Reviewer_nHXh · 2024-11-02

**Soundness:** 3
**Presentation:** 3
**Contribution:** 3
**Rating:** 6
**Confidence:** 3

**Summary:**

The paper describes an approach for improving the logical consistency and factuality of language models, using neuro-symbolic integration.
The paper starts with a list of facts and logical constraints. All the valid combinations of truth values for these facts are then iterated and used as targets during optimization.
The experiments evaluating the correctness and consistency of the learned facts show that this method outperforms vanilla models and a baseline using an external solver.

**Strengths:**

The paper is making advancements in neurosymbolic modelling.
It is certainly a nice achievent to not have to rely on an external solver and being able to push the knowledge into the main neural model.

**Weaknesses:**

The evaluation is the weakpoint of the paper at the moment.

Macaw-Large, which is used for the main experiments, is quite old already (pre-LLM).
Even Llama-2 used in later experiments is much less capable on most tasks compared to the current Llama-3.2.
This raises questions how applicable the proposed methods are to the current generation of language models.

The main baseline is CONCORD, which is from 2019 and uses RoBERTa.
The fact that the proposed system is able to outperform this baseline without using an external solver is great.
But there really should be some additional baselines with newer methods that also use model updating.
For example, there is a whole library of papers focussing on updating specific facts in language models using targeted fine-tuning.

The whole evaluation is performed on very artificial tasks. It would be very useful to see how these changes impact the model performance in practical applications.


Asking the LLM “Is an albatross not an organism?” is a very unnatural phrasing, whereas LMs are trained to predict natural continuations. I suspect that may be negatively affecting the performance for LMs.

**Questions:**

The method relies on collecting the probabilities for specific tokens to estimate the yes/no probabilties.
How much is this going to be affected by the label bias of the LLMs?
https://openreview.net/forum?id=shr9PXz7T0
https://arxiv.org/pdf/2402.09910

---

> ### Author Response · Authors · 2024-11-22
>
> We thank the reviewer for the feedback and for appreciating how our approach allows us to side-step external solvers.  We address below all the concerns they raised.
>
> > *Macaw-Large [...] is quite old already.  Even Llama-2 [...] is much less capable [...] compared to the current Llama-3.2. This raises questions [of applicability].*
>
> This was a forced choice to enable comparison against ConCord, which relies on Macaw and does not scale to newer and larger LLMs.
> In Table 2 we now also report the performance of Llama3.1 8B and show that, while the model is supposed to be more capable at reasoning than Llama 2, it falls short on BeliefBank in the very same way Llama2 7B with Few Shot. We are in the process of running the finetuning of LoCo-Llama3.1 and expect it to show similar improvements under different constraints.
>
> > *there really should be some additional baselines with newer methods that also use model updating. For example, there is a whole library of papers focussing on updating specific facts in language models using targeted fine-tuning.*
>
> We’d be glad to compare against additional baselines, provided our computational budget allows it.  What other approaches do you think we should consider?
>
> > *It would be very useful to see how these changes impact the model performance in practical applications.*
>
> We are open to run additional experiments if the reviewer provides concrete suggestions for concrete applications that we can execute during the rebuttal.
>
> > *Asking the LLM “Is an albatross not an organism?” is a very unnatural phrasing, whereas LMs are trained to predict natural continuations. I suspect that may be negatively affecting the performance for LMs.*
>
> We’d be glad to test additional prompts.  Please let us know what you think we should test.
> We have updated our scores in Table 2 (and Tables 6-14 in the Appendix) with other syntactical variations of prompts, see also the new Appendix F and our answer to reviewer jLBW.
>
> > The method relies on collecting the probabilities for specific tokens to estimate the yes/no probabilties. How much is this going to be affected by the label bias of the LLMs? https://openreview.net/forum?id=shr9PXz7T0 https://arxiv.org/pdf/2402.09910
>
> That’s a good point, but our LoCo-LMs are as susceptible as other LLMs to this phenomenon. We note that there is no a priori selection bias (as referred in the paper you linked) in the constraints defined in BeliefBank and EntailmentBank, therefore we do not believe that the semantic loss is affected in this sense.

---

> > ### Comment · Reviewer_nHXh · 2024-11-26
> >
> > Thank you for your response. I will keep my original assessment.
> >
> > Sorry, but as I am not a co-author on this paper, I am not able to put together a detailed step-by-step guide for how to best address each of the shortcomings.

---

### Official Review · Reviewer_g2CN · 2024-11-03

**Soundness:** 3
**Presentation:** 2
**Contribution:** 3
**Rating:** 6
**Confidence:** 3

**Summary:**

This work proposes a fine tuning method for improving logical consistency in language models. Given a set of facts and a set of constraints the idea is to finetune the model to make sure that certain logical constraints are respected, typically implication and negation. The authors show that indeed finetuning allows to improve self consistency and that this transfers beyond the facts and constraints used for finetuning to other entities and and settings.

**Strengths:**

The problem of improving logical consistency in language models is important. The approach is simple and does not require a lot of inference time compute since it is based on finetuning. The empirical results that show transfer and generalization beyond the training distribution are informative and interesting.

**Weaknesses:**

* Clarity: the paper can do a better job at explaining the details of its method. The authors spend two pages (section 2) on explaining logical constraints in a way that is too elaborate (for example, defining the xor operator in line 124, and defining implication in terms of negation and or in line 137) and unnecessary. On the other hand details on the actual method is limited (see questions below), specifically the paragraph in 232 and the precise process of how logical constraints are transformed into differentiable graphs are explained in a manner that is insufficient. The description of the experiments also mixes unimportant implementation details with more important details on the experimental setup which makes it hard to understand the details of the experiments and what can be concluded from them.

* Related to the above - Figure 1 takes a lot of real-estate but is not helpful. The only thing we see is that there is baseline that makes a mistake on 3 examples and the proposed model does not make the mistake. This does not say a lot on the method, or the aggregate results only we can learn about the types of logical constraints that will be used. This might be ok if the important parts of the paper were clear, but they are not sufficiently clear at this point.

* Key point that was unclear to me:  line 242 paragraph: I don’t understand if the method handles facts that can be inferred from \alpha and the KB but require more than one hop? When training the SL loss, are those considered? Say we have in the KB, “albatross is a bird”, “birds are an animal”, “albatross can fly”, “if an animal can fly then the animal can move”. Will the SL loss contain a term about whether albatrosses and whether they can move or not? Is this done implicitly somehow? Where do we do the inference of all potential things that can be inferred from the KB and the constraints and take those into account in the SL loss?

* More on clarity: in section 3, you define \mathcal{D}_c = \{alpha_1, \dots, \alpha_m\}. But the structure of \alpha is not clearl defined. It would be gold ot make this much clearer, it becomes clearer later as you read more, but should be explained better at this point.

* Clarity: z ~p_\theta(z) is confusing. Supposedly p_theta is the language model and it look like sampling from the unconditional distribution of text, but the text says something else, that it is sampling truth assignments conditioned on what appears in \alpha_i but this is not clear from the notation.

Another key point are some problems with clarity and worries about the experimental setup.

* IIUC the only baseline that is reported that is not from the authors is ConCord for which two numbers exactly are reported and that's it. There is some reference to maieutic prompting but it is unclear if this should be another baseline or is too similar to ConCord. It is not clear if there are not reasonable baselines to compare other than that. There is reference to few-shot baselines, but it is not expalined what are the examples in the few-shot examples and how they are supposed to help, in fact in many cases results are worse for few-shot compared to zero-shot. Overall, the authors should make clear if there is no past work beyond ConCord and just finetuning on the KB (XENT) without using the constraints

* Second, for ConCord, it seems that the authors use ROBERTA-ANLI as an inference model. But for their LOCO method it seems like they are using hard constraints that are guaranteed to be true - if that's the case this is unfair towards ConCORD. Can the authors provide more details about how and why they outperform ConCord? Do the two methods use the same models and same constraints? Form the fact that the authors say that Concord requires ROBERTA-ANLI it sounds like the answer is "no" but would be good to understand better what's going on. Since we only have two numbers in the paper that are not baselines implemented by the authors it is important to understand the details in this setup.

To conclude, I found the overall premise of the paper interesting but the paper needs to be clearer both in terms of method and in terms of experimental results and how they relate to past work.

**Questions:**

* Line 192: the authors claim that they expect transfer from albatross to cockerel since they are similar - but there is no definition of what is similarity, and how should the model know when things are similar enough to conclude new facts about entities and when not. I assume this refers to some vague simlarity measure in the space of hidden representations, but this is still confusing.

* Line 469 - where are the resutls? are they in Table 3? the paper doesn't say

* What is the few-shot baselines precisely? what are the examples given and how are they helpful?

---

> ### Author Response · Authors · 2024-11-21
>
> We thank the reviewer for the feedback and for appreciating that our approach tackles an important problem and that, while simple and efficient, yields benefits beyond the training set.  We address below all the concerns they raised.
>
> > *[too much space] explaining logical constraints*
>
> Our goal was that of equipping NLP researchers, which might not be overly familiar with the topic, with the necessary preliminaries.  We find this is essential for understanding LoCo-LMs and the problem they aim to solve: In our interactions with NLP researchers, that was the hardest part to understand for them.
>
> > *details on the actual method is limited*
>
> Thank you for pointing this out.  We have added an overview of the overall pipeline in Figure 1 and detailed how the circuit is constructed in Appendix A.
>
> Please let us know if there are any other details that you find are unclear, we’ll be glad to detail them further in the revised manuscript.
>
> > *the experiments mix unimportant and important details*
>
> We are happy to restructure it, if you can specify which details you would prefer to be moved to the appendix.
>
> > *line 242: I don’t understand if the method handles facts that can be inferred from \alpha and the KB but require more than one hop?*
>
> The loss enforces constraints on given facts from a fixed-size knowledge base. We are not proposing a way to augment knowledge bases via deduction (i.e., generating new facts).
>
> That being said, if we are given a multi-hop reasoning constraint (see our EntailmentBank experiments and Appendix D), we can enforce logical consistency over multiple reasoning steps. I.e., the formula $\alpha$ can reference arbitrarily many given logical facts; the semantic loss term into which $\alpha$ is compiled will constrain exactly those facts.  E.g., for $\alpha$ is “(albatross is a bird) AND (birds are animals) => (albatross are animals)” the value of the SL depends on the probability that the model assigns to all these facts holding.
>
> > *section 3: \mathcal{D}_c = {alpha_1, \dots, \alpha_m}, but the structure of \alpha is not clearly defined.*
>
> $\alpha$ refers to formulas like Eq. (Imp), (Neg), (F-Imp) and (2) in Section 2.  We had mentioned this in line 186.  We have made this more explicit.
>
> > *z ~p_\theta(z) is confusing.*
>
> The individual probabilities $p_\theta(z)$ are obtained using Eq. (1).  We have added a backref in the text.
>
> We stress that in LoCo-LMs the circuit computes the probability that a constraint holds for the model (i.e., that expectation) **exactly**,  no sampling is required.
>
> > *the only baseline is ConCord.*
>
> We found only ConCord as a sensible baseline and we point out that maieutic prompting is just a variant of ConCord. In fact, maieutic prompting implements the same algorithm: exactly like ConCord, it extracts raw fact probabilities from the LLM and refines them so as to be as consistent to each other as possible using a MAX-SAT solver. We have clarified this in line 317.
> Furthermore, besides ConCord, we compare also against Chain-of-Thought (Section 5.2).
>
> > *what are the examples in the few-shot examples?*
>
> Thanks for pointing this out, we added them to Appendix F.3.
>
> > *For ConCord, it seems that the authors use ROBERTA-ANLI as an inference model. [...] this is unfair towards ConCORD. Do the two methods use the same models and same constraints?*
>
> We use ConCORD as originally proposed, and we use the same MACAW models. Hard constraints are enforced by a MaxSAT solver, therefore they are guaranteed to hold ultimately.
>
> The difference is that ConCORD uses ROBERTA-ANLI to *propose* facts to ground the constraints, while we just maximise the probability of the constraints given the data.  At inference time, LoCo-LMs do not guarantee that constraints are satisfied.
>
> Therefore, one could use LoCo-LMs as a loss at training time and then combine it with a MaxSAT solver at inference time.
>
> The difference in performace reflects a difference in approach: ConCORD attempts to rectify post-hoc the predictions of a potentially non-factual, inconsistent model, but this at best can help with self-consistency, not with factuality.  LoCo-LMs on the other hand make the model and its answers both more self-consistent and factual.
>
> > *Line 192: the authors claim that they expect transfer from albatross to cockerel since they are similar - but there is no definition of what is similarity, and how should the model know when things are similar enough to conclude new facts about entities and when not.
>
> The idea is that entities that map to similar embeddings (via, e.g., cosine similarity) will yield similar activations.  Hence, applying the SL to one entity is likely to yield benefits for entities similar to it.  Empirically, this is what happens in our tests.  We were careful not to claim this will happen with guarantees
>
> > *Line 469 - where are the resutls? are they in Table 3?*
>
> Correct, we amended the text.

---

> > ### Comment · Reviewer_g2CN · 2024-11-23
> > **Thanks for the response, a few more questions**
> >
> > * Regarding comparison with ConCord and use of ROBERTA. I still don't fully understand this. (a) You say that you use ROBERTA-ANLI to propose grounded facts. But it seems that LOCO LM also requires a step of grounding abstract inference rules, which is done by matching subjects (line 316). So what is the difference? Why is there a need for ROBERTA in one but not in the other, even if that is how it was done originally. (b) You explain concord can help with self-consistency, but not with factuality. But it seems possible to do this with a variant where you train the model with XENT to have high probability on the true facts and use the SAT solver for consistency? Even if these things haven't been done in the past, they seem important for the claim that you get better performance by fine-tuning for consistency and factuality rather than using a solver.
> >
> > * I understand know that there is no deduction proposed. So IIUC if we don't assume generalization across entities (like in sections 5.3 and 5.4) then we only expect to get consistency w.r.t what the KB actually contains. Isn't this a limitation on generality? you need to explicitly have in the KB all of the facts for which you hope to achieve for all of the entities (millions potentially?), and you need to ground all of them with all of the abstract inference rules leading to an explosion of terms in the loss function. If that is true then the method is general only as long as there is good transfer across even unrelated entities. Sections 5.3 and 5.4 give some results but it is hard to understand if this procedure will lead to noticeable differences when the coverage of the KB is limited.
> >
> > * Notation of p_theta(z): in line 183 you say p_theta encodes a distribution over tokens. But then in equation 3 it is used to mean something else related to definitions in a previous section. I find this notation to be confusing and should be improved.
> >
> > * Regarding experimental parts that can be moved to appendix. I propose as examples lines 341-344. Also in a similar fashion details in the paragraph of line 384
> >
> > * The authors say the process doesn't hurt fluency, but seems like perplexity does meaningfully go up.
> >
> > I am still raising my score, thanks for the response

---

> > > ### Author Response · Authors · 2024-11-24
> > > **Thanks for the prompt response**
> > >
> > > Thank you for your follow-up. We gladly clarify your doubts as follows, hoping for a full acceptance.
> > >
> > > > *You say that you use ROBERTA-ANLI to propose grounded facts. But it seems that LOCO LM also requires a step of grounding abstract inference rules, which is done by matching subjects (line 316). So what is the difference?*
> > >
> > > There is a misconception here that makes ConCoRD and LoCo-LM not directly comparable: LoCo-LM operates at training time, where ground truth grounded constraints (from the training set) are available, while ConCoRD operates at test time, where there are not already-grounded constraints available.
> > >
> > > As such, LoCo-LM can exploit the information of ground truth constraints in the very same way that finetuning with cross-entropy (XENT) does at training time, but not at test time. For this same reason, we very carefully evaluate train/valid/test splits by making sure that there is no leak of ground entities between sets (see our T1 vs T2 splits, and our answer later for generalization).
> > >
> > > ConCoRD on the other hand, uses ROBERTA-ANLI to extract relationships among the queried facts at test time, as no ground truth is available (also for LoCo-LM) by then. As we said, one could combine the two techniques and have a LoCo finetuning at training time and then enhance consistency of extracted facts and relationships with a MaxSAT solver at test time.
> > >
> > > > *if we don't assume generalization across entities (like in sections 5.3 and 5.4) then we only expect to get consistency w.r.t what the KB actually contains. Isn't this a limitation on generality?*
> > >
> > > We do not understand how this can be a reason to reject the paper as that would be the expected behaviour for *any logic reasoner*, if one cannot assume generalisation across entities. So while it is a limitation, it is a limitation of all logical reasoning. We stress that this does not apply as, thanks to operating with an LLM, we are able to generalize to unseen KBs and other concepts that are semantically related (but syntactically different) as you already point to Sections 5.3 and 5.4.
> > >
> > > > *you need to explicitly have in the KB all of the facts for which you hope to achieve for all of the entities (millions potentially?), and you need to ground all of them with all of the abstract inference rules leading to an explosion of terms in the loss function.*
> > >
> > > This is not true, as a modest semantic overlap can already provide enough mileage as shown in section 5.3 and 5.4. See also our heatmap in Figure 2 in the Appendix where 7 entities are enough to help boosting (self-)consistency for 80+ new entities. Furthermore, we do not see finetuning on a large KB (there are plenty, see WikiData) to be an inherent problem if someone had the resources to do so.
> > >
> > > > *Notation of p_theta(z): in line 183 you say p_theta encodes a distribution over tokens. But then in equation 3 it is used to mean something else related to definitions in a previous section. I find this notation to be confusing and should be improved.*
> > >
> > > $p_{\theta}(\mathbf{x})$ is a distribution over tokens, which induces a distribution over fact truth values $p_{\theta}({z})$ see Eq 1. We will rephrase line 183 to make this clear.
> > >
> > > > *Regarding experimental parts that can be moved to appendix. I propose as examples lines 341-344. Also in a similar fashion details in the paragraph of line 384*
> > >
> > > Thanks for the suggestion, we will move them in the next version.
> > >
> > > > *The authors say the process doesn't hurt fluency, but seems like perplexity does meaningfully go up.*
> > >
> > > The rise in perplexity from ~52 to ~62 can be explained by the fact that our finetuned models are all quantized 4 bits, while the reported baselines are unquantized. A quantized vanilla Llama scores ~62 perplexity. We will underline this in the next revision.
> > >
> > > We are happy to answer any further doubt left.

---

> > > > ### Comment · Reviewer_g2CN · 2024-11-25
> > > > **Thanks for the additional explanations**
> > > >
> > > > I appreciate the additional explanations.
> > > >
> > > > * I still fail to understand the mismatch between concord and loco-lm. IIUC, the inference rules that are used for generating training constraints are *abstract*. I don't really see in what sense they cannot be used at test time. given some query about a fact, I can generate additional facts by instantiating these abstract inference rules and choose the assignment that maximizes probability while respecting the hard constraint. Are you saying this is impossible? or this is cheating? or would lead to worse performance? I don't see why but could be wrong.
> > > >
> > > > Regardless I agree it's valuable to see how fine-tuning compares to post-hoc constraint enforcing with a max-sat solver but it'd be good to understand if we can make the setups as close as possible.
> > > >
> > > > * About the generalization point, you might be right that this is something that might apply more broadly to additional papers. I think that if you can only achieve consistency w.r.t to a KB without any deduction, just applying manually-written constraints to manually-specified facts, then using KBs for enforcing consistency is probably of too limited generality. Feel free to reach out to the AC if you think this is an unreasonable position. So yes, I think generalization is key in this case and the results definitely seem encouraging in 5.3 but more brittle in 5.4. I would be surprised if using KBs for improving LLMs consistency will become common if consistency is w.r.t to KB facts and constraints only. Can you provide applications where KBs with facts and constraints are sufficient to achieve broad consistency without deduction and when OOD generalization results are mixed?

---

> > > > > ### Author Response · Authors · 2024-11-26
> > > > > **Thanks for following up**
> > > > >
> > > > > Thanks for the quick follow-up, very appreciated! We answer in the following.
> > > > >
> > > > > > *I still fail to understand the mismatch between concord and loco-lm. IIUC, the inference rules that are used for generating training constraints are abstract. I don't really see in what sense they cannot be used at test time*
> > > > >
> > > > > This is true, and that’s exactly what ConCoRD does: it uses  ROBERTA-ANLI to instantiate the rules and get grounded constraints. Then a MaxSAT solver comes up with the (truth values of) facts that maximise the probability. We refer the reviewer to Figure 2 in the ConCoRD paper.
> > > > >
> > > > > As such, ConCoRD is already using all possible information at test time. LoCo-LMs instead use the information at training time. Combining them is an interesting future line.
> > > > >
> > > > > >  *I think that if you can only achieve consistency w.r.t to a KB without any deduction, just applying manually-written constraints to manually-specified facts, then using KBs for enforcing consistency is probably of too limited generality.*
> > > > >
> > > > > We do not see this as a limitation (that can kill a paper!), in the sense that logical constraints are always assumed to be given (Abstract constraints are given in ConCoRD, see comment above). And where there are none, one can always learn them from data [A, B] and later apply LoCo-LM as a subroutine in a larger loop where constraints are refined. This is an interesting future-work perspective, that would need LoCo-LM to be established.
> > > > >
> > > > > >  *Feel free to reach out to the AC if you think this is an unreasonable position.*
> > > > >
> > > > > As the discussion so far has been polite and fruitful, we do not think we should contact the AC : )
> > > > > We hope we can keep discussing it as to clarify doubts. We remark that there are many interesting open research questions that LoCo-LM can enable, but ***solving all of them now does not fit a single paper***.
> > > > >
> > > > > >  *I think generalization is key in this case and the results definitely seem encouraging in 5.3 but more brittle in 5.4.*
> > > > >
> > > > > We find them both promising, and we remark that these kinds of “out-of-distribution” generalization problems have not been touched in previous works, e.g., ConCoRD. Also there generalization is bounded by the (implicit) knowledge in ROBERTA-ANLI and the given constraints. There is no guarantee that using another NLI LLM the MaxSAT solution would be anywhere similar.
> > > > >
> > > > > > *Can you provide applications where KBs with facts and constraints are sufficient to achieve broad consistency without deduction*
> > > > >
> > > > > Could you please elaborate further? Our experiments on BeliefBank, EntailmentBank and ConceptNet are exactly doing this. If you want broader pointers to a literature outside NLP, we refer you to the neurosymbolic literature [C, D], where constraints and KBs are coming from experts and do not change so frequently with time.
> > > > >
> > > > > [A] De Raedt, Luc, Andrea Passerini, and Stefano Teso. "Learning constraints from examples." Proceedings of the AAAI conference on artificial intelligence.2018.
> > > > >
> > > > > [B] Bessiere, Christian, et al. "Constraint acquisition." Artificial Intelligence 244 (2017): 315-342.
> > > > >
> > > > > [C] Giunchiglia, Eleonora, et al. "CCN+: A neuro-symbolic framework for deep learning with requirements." International Journal of Approximate Reasoning (2024): 109124.
> > > > >
> > > > > [D] Ahmed, Kareem, et al. "Semantic probabilistic layers for neuro-symbolic learning." Advances in Neural Information Processing Systems 35 (2022): 29944-29959.

---

> > > > > > ### Comment · Reviewer_g2CN · 2024-12-02
> > > > > > **One last note**
> > > > > >
> > > > > > Hi,
> > > > > > I am not sure I got the roberta-anli bit 100% but that's ok for now.
> > > > > >
> > > > > > I will raise the score since I think the technical approach makes sense and it is worth applying large language models for better consistency. The experimental investigation also seems solid. I remain skeptical whether at a high level this approach will scale and we can automatically extract rules and then train for self-consistency w.r.t to those rules in a way that will be generally useful, but I agree with the authors this is a reasonable direction that is worth pursuing and seeing how far it can go.

---

### Official Review · Reviewer_hYcP · 2024-11-04

**Soundness:** 4
**Presentation:** 3
**Contribution:** 3
**Rating:** 8
**Confidence:** 3

**Summary:**

This paper introduces LoCo-LLM, a fine-tuning method for LLMs that leverages a neuro-symbolic inspired semantic loss function to enhance its factuality and logical consistency. The proposed semantic loss function is based on weighted model counting, with weights derived from the LLM’s probability estimates. LoCo-LLM employs sentential decision diagrams to efficiently compute this loss.

Detailed experiments compare LoCo-LLM with baselines that use external reasoners and traditional cross-entropy-based fine-tuning. Experimental results on the BeliefBank and EntailmentBank datasets show that the proposed framework outperforms baselines on metrics such as factuality and consistency.

The code to reproduce these results is provided as supplementary material and will be released on GitHub under a permissible license.

**Strengths:**

- The idea of using a neuro-symbolic loss function to improve logical consistency and factuality in LLM responses is novel and interesting. The proposed loss function is generalizable, can be extended to complex logical constraints, and may prove useful in enhancing LLMs' reasoning capabilities.
- The detailed experimental results demonstrate the advantages of the proposed method over baselines, even on relatively small (5-10%) datasets.

**Weaknesses:**

- Although the loss function is explained thoroughly, other components, such as circuits and sentential decision diagrams, are not discussed in detail. Including these details would improve the paper's readability.

- The experiments are conducted on datasets with outputs of fewer than 4 tokens, leaving it unclear how well the proposed method supports generating longer, factually and logically consistent responses.

**Questions:**

- For the pre-trained baseline models in Tables 1 and 2, do the scores improve with greedy decoding?

---

> ### Author Response · Authors · 2024-11-22
>
> We thank the reviewer for the feedback and for appreciating that our work is novel and interesting, that our approach can handle complex constraints and is empirically promising.  We address below all the concerns they raised.
>
> > *other components, such as circuits and sentential decision diagrams, are not discussed in detail*
>
> Thank you for pointing this out, we (wrongly) assumed that the theory of knowledge compilation was consolidated in the neuro-symbolic community.
> We introduced a new Appendix A to revise the background on circuits and compilation.
>
> Note that for our experiments, we use standard compilation tools from the knowledge compilation literature to obtain a circuit starting from a propositional logical formula in conjunctive normal form. Specifically, we use PySDD2 [x], a python compiler that converts logical formulas into Sentential Decision Diagrams (SDDs) [y, z].
> For example, given a formula such as (albatross => bird), the compiler instantiates two nodes for each variable encoding, in this case, whether albatross holds, albatross does not hold, bird holds, and bird does not hold, respectively. These nodes store the probabilities of these events. The compiler then adds sum and product nodes – which, very intuitively, compute the sum and product of their inputs – to the SDD, which is structured such that bottom-up evaluation of the circuit yields the probability that the formula holds given the probabilities of the input events.
> A more detailed step-by-step example is shown in Appendix A.
>
> [x] (2017). Pysdd. In Recent Trends in Knowledge Compilation, Report from Dagstuhl Seminar 17381.
>
> [y] Choi, A. and Darwiche, A. (2013). Dynamic minimization of sentential decision diagrams. AAAI.
>
> [z] Darwiche, A. (2011). SDD: A new canonical representation of propositional knowledge bases. IJCAI.
>
> > *4 tokens max; unclear how well the proposed method supports generating longer [...] responses.*
>
> We would like to point out that there is a difference between the number of logical variables appearing in a formula and the number of tokens produced by the model.
>
> The former ranges from a minimum of 1 to $2^D$ where $D$ is the depth of the implication trees in EntailmentBank [Dalvi et al., 2022].  Please see Figure 2 in the Appendix for one example.
>
> The number of tokens used to evaluate the probability of every fact is instead 1. See also our prompts in Appendix F.
>
> [Dalvi et al., 2022] Dalvi et al. Explaining answers with entailment trees. EMNLP 2022.
>
> > *do the scores of baselines in Tables 1 and 2 improve with greedy decoding?*
>
> In Tables 1 and 2, we used the default decoding strategy for Llama.  We re-run our evaluation on LoCo-SUPER using greedy decoding, and found that performance is essentially the same. We reported the scores for all constraints in Table 18 in the Appendix.

---

### Official Review · Reviewer_jLBW · 2024-11-05

**Soundness:** 3
**Presentation:** 3
**Contribution:** 2
**Rating:** 6
**Confidence:** 4

**Summary:**

The paper explores improving LLMs' factuality and logical consistency through neuro-symbolic reasoning. It introduces a neuro-symbolic loss function that is used to fine-tune LLMs on a given set of external facts and rules. Experiments show that this approach achieves improved consistency and generalizes more effectively to unseen yet similar constraints compared to baseline methods, including those that rely on external reasoning tools.

**Strengths:**

The paper offers a novel approach by integrating neuro-symbolic reasoning into the fine-tuning of large language models (LLMs) to improve factuality and logical consistency. While existing approaches for enhancing consistency in LLMs often rely on external reasoning tools or extensive fine-tuning, this paper proposes a middle-ground solution: a neuro-symbolic-based loss function that promotes logical consistency by maximizing the probability of constraint satisfaction. This approach (LoCo-LMs) is grounded in weighted model counting and semantic loss, offering a flexible framework that applies consistently across various logical constraints, such as negation and implication.

The paper conducts extensive experiments to showcase LoCo-LMs' effectiveness over traditional approaches, demonstrating improvements in logical consistency, factuality, and transferability across different logical constraints and datasets. The method also proves efficient, achieving good performance even with limited training data.

By enhancing logical consistency without requiring external reasoning frameworks, the approach has important implications for deploying LLMs in tasks that demand reliable, logic-based reasoning. Its ability to generalize to unseen (yet semantically similar) facts presents a promising pathway for real-world applications where models need to work reliably with sparse data.

**Weaknesses:**

Evaluation scope:

The experiments primarily focus on logical constraints such as negation, implication, and reverse implication. While these are fundamental, they fall short of capturing the more complex reasoning scenarios often required in real-world applications. For instance, the paper could improve by incorporating evaluations on multi-hop reasoning tasks or exploring more sophisticated logical constraints.


Shift in language modeling distribution:

The authors assess possible shifts in the language modeling distribution by measuring changes in perplexity, yet their evaluation could be expanded. Adding downstream tasks (e.g, question answering, reading comprehension, mathematical reasoning, etc.) would allow to assess whether the proposed fine-tuning approach not only improves logical consistency but also maintains the language capabilities of the original model.


Robustness of the results:

The experiments reveal that fine-tuning LoCo-LMs improves generalization only within the same type of constraints, and it even hurts performance when the constraints differ between fine-tuning and testing (see Table 4). This limitation could be especially pronounced in smaller models, so testing on larger models could provide further insights. It would also be valuable to explore whether these performance gains also transfer to more capable models, such as comparing performance between LlaMa 2 and LLaMa 3, with and without LoCo-LMs.


Sensitivity to prompting:

The effectiveness of the approach appears to be sensitive to the specific prompt formats used during fine-tuning and evaluation. This suggests that the gains in consistency might be partially due to prompt selection rather than the model’s inherent logical coherence. Broader testing across diverse prompt templates would enhance the robustness and reproducibility of the results. Moreover, there are alternative prompting methods to elicit logical consistency, such as prompting the model to respond sequentially to a series of related questions, conditioned on previous answers.

**Questions:**

Please see "Weaknesses" section.

---

> ### Author Response · Authors · 2024-11-22
>
> We thank the reviewer for the feedback and for appreciating that our work is novel and that our approach is flexible, empirically promising and significant for applications.  We answer  below all the questions they asked.
>
> > *the paper could improve by incorporating evaluations on multi-hop reasoning tasks / more sophisticated logical constraints.*
>
> We remark we already used constraints involving more than one implication: This analysis can be found in Section 5.3, where we evaluated LoCo-LMs on EntailmentBank.  This consists of entailment **trees** involving multiple inference steps across multiple entities/logical variables; see Appendix D, Figure 2 for a visualization for an implication tree.  The number of steps ranges from 1 to 5, see Figure 2 of [Dalvi et al., 2022] for the precise distribution.
>
> We have made sure to clarify this point at  the beginning of Section 5.4.
> We are happy to discuss this further.
>
> [Dalvi et al., 2022] Dalvi et al. Explaining answers with entailment trees. EMNLP 2022.
>
> > *Adding downstream tasks (e.g, question answering, reading comprehension, mathematical reasoning, etc.) [...] to assess whether the proposed fine-tuning approach [...] maintains the language capabilities of the original model.*
>
> This is a good idea! Unfortunately, our computational resources are limited and we might not be able to provide results for these additional tasks during the discussion period.  We will try our best to integrate a such an evaluation.
>
> > *improves generalization only within the same type of constraints, and it even hurts performance when the constraints differ*
>
> We remark this is expected and common when doing multi-objective optimization.
> Optimizing one constraint might not always benefit all others as much as it benefits its own kind.
>
> Note that, however, the great majority are cases of positive transfer, i.e., optimizing for one constraint also benefits others. For example, optimizing for NEG improves all columns of Table 2 wrt the baseline (C-FAC: +19%, C-IMP: +20%, C-REV: +42%, SC-REV: +35%) but self-consistency IMP, and optimizing F-IMP only degrades self-consistency REV and NEG (C-FAC: +74%, C-REV: +8%), as it rightly does not consider negation, while delivering much better performance over all cases than using XENT. We will highlight these relative improvements in the Table for the camera-ready version.
>
> We have integrated this discussion in the paper in Section 5.2.
>
> > *This limitation could be especially pronounced in smaller models, so testing on larger models [is warranted] (e.g., llama 2 vs llama 3).*
>
> In Table 2 we also report the performance of Llama3.1 8B and show that, while the model is supposed to be more capable at reasoning, it falls short on BeliefBank as Llama2 7B with Few Shot. We are in the process of running the finetuning of LoCo-Llama3.1 and expect it to show similar improvements under different constraints.
>
> > *gains in consistency might be partially due to prompt selection -> test other prompts or prompting methods*
>
> This is a good question. We note that we already performed experiments on two prompts (as shown in Appendix F). We have now extended our analysis to two more prompts (using as syntactic variations `correct`/`incorrect` and `right`/`wrong`).
>
> We can observe that the performance we previously reported remains stable and our claims still hold: LoCo-LMs improve consistency for the different constraints even on alternative prompts.

---

> > ### Comment · Reviewer_jLBW · 2024-11-27
> >
> > Thank you for addressing the questions and providing clarifications. I appreciate the effort you put into revising the manuscript, as these updates should also benefit potential readers of the paper.

---

> > > ### Author Response · Authors · 2024-11-27
> > > **Thanks!**
> > >
> > > Thanks for the appreciation and for engaging with us. Would you mind reflecting it in an updated score?

---

### Public Comment · ~Wen-Da_Wei1 · 2024-11-14
**Minor questions about the paper**

I have some minor questions about the content of the article. I hope the author can help me resolve my doubts, and thank you in advance.
How can the method proposed ensure that a model fine-tuned solely with semantic loss achieves self-consistency and other reasoning capabilities which most large language models cannot reach?
I think this method can only ensure the model is factual, and it's hard to achieve other effects.

---

### Author Response · Authors · 2024-11-28
**Thank you & additions to our submission**

We thank all reviewers for their insightful feedback, questions, and kind words. We are glad that the paper has been appreciated for **tackling an important task** (“has important implications for deploying LLMs in tasks that demand reliable, logic-based reasoning”, jLBW, “improving logical consistency in language models is important”, g2CN, “certainly a nice achievement to not have to rely on an external solver”, nHXh) being **theoretically rigorous** (“ a fine-tuning method grounded in neural-symbolic reasoning”, G8eP) and **a novel approach** (“  using a neuro-symbolic loss function(...) is novel and interesting”, hYcP, “a novel approach by integrating neuro-symbolic reasoning”, jLBW) that is **effectively validated empirically** (“detailed experimental results”, hYcP, “extensive experiments to showcase LoCo-LMs' effectiveness over traditional approaches”, jLB).
We answered all the concerns raised by each reviewer below. We highlight that **we added additional baselines** and we **fine-tuned an updated architecture** (LLaMa 3.1 8b, see our response to jLBW, our claims still hold); **we introduced new prompt formats** (found consistency with the previous scores, updated the tables, see our response to jLBW, nHXh); we **tested an alternative decoding strategy** (Greedy decoding, in response to hYcP); **we introduced a background section on how to compile logical formulas into computational graphs** (Appendix A, in response to hYcP)

Please let us know if there are some aspects you would like to discuss more. We are keen on engaging during rebuttal towards a full acceptance of this paper.

---

### Meta-Review · Area_Chair_CgPZ · 2024-12-23

**Metareview:**

The paper proposes LoCo-LMs, a neuro-symbolic fine-tuning method using semantic loss to enhance LLMs’ logical consistency and factuality. It reduces reliance on external tools and shows improved consistency and generalization over baselines. Strengths include novelty, efficiency, and empirical validation. Weaknesses are limited evaluation scope, reliance on older models, and sparse comparisons to modern baselines. Decision: marginal acceptance.

**Additional Comments On Reviewer Discussion:**

The rebuttal addressed key concerns by adding baselines, refining evaluations, and clarifying methods. Reviewers acknowledged improved generalization and practical relevance but noted limitations in scalability and downstream evaluations. The authors’ responses strengthened the case for marginal acceptance.

---

### Decision · Program_Chairs · 2025-01-22

Accept (Poster)